# Missense mutations in CRX homeodomain cause dominant retinopathies through two distinct mechanisms

**Yiqiao Zheng[1,2], Chi Sun[1,2], Xiaodong Zhang[2], Philip A Ruzycki[2,3], Shiming Chen[1,2,4]\***

[1]Molecular Genetic and Genomics Graduate Program, Division of Biological and Biomedical Sciences, Washington University in St Louis, Saint Louis, United States; [2]Department of Ophthalmology and Visual Sciences, Washington University in St Louis, Saint Louis, United States; [3]Department of Genetics, Washington University in St Louis, Saint Louis, United States; [4]Department of Developmental Biology, Washington University in St Louis, Saint Louis, United States

**\*For correspondence:**
chenshiming@wustl.edu

**Competing interest:** The authors declare that no competing interests exist.

**Abstract** Homeodomain transcription factors (HD TFs) are instrumental to vertebrate development. Mutations in HD TFs have been linked to human diseases, but their pathogenic mechanisms remain elusive. Here, we use *Cone-Rod Homeobox* (*CRX*) as a model to decipher the disease-causing mechanisms of two HD mutations, p.E80A and p.K88N, that produce severe dominant retinopathies. Through integrated analysis of molecular and functional evidence in vitro and in knock-in mouse models, we uncover two novel gain-of-function mechanisms: p.E80A increases CRX-mediated transactivation of canonical CRX target genes in developing photoreceptors; p.K88N alters CRX DNA-binding specificity resulting in binding at ectopic sites and severe perturbation of CRX target gene expression. Both mechanisms produce novel retinal morphological defects and hinder photoreceptor maturation distinct from loss-of-function models. This study reveals the distinct roles of E80 and K88 residues in CRX HD regulatory functions and emphasizes the importance of transcriptional precision in normal development.

## eLife assessment

This manuscript will be of interest to readers in the field of neural development and neurodegeneration. The study is **important** as it examines two disease-causing mutations within the homeodomain transcription factor Cone-Rod Homeobox (CRX) that causes retinopathy in humans. The data are **solid**, and the work contributes to our understanding of the underlying pathogenetic mechanisms.

## Introduction

Homeodomain transcription factors (HD TFs) play a fundamental role in vertebrate development. Members of the HD TF family are characterized by the presence of a highly conserved 60 amino acid helix-turn-helix DNA-binding domain known as the homeodomain (HD). The HD is one of the most studied eukaryotic DNA-binding motifs since its discovery in *Drosophila* homeotic transformations (*Mark et al., 1997*; *Lewis, 1978*). Hundreds of HD TFs have subsequently been documented in regulating gene expression programs important for body plan specification, pattern formation, and cell fate determination (*Mark et al., 1997*; *Lewis, 1978*). Mutations in HD TFs have been linked to many human diseases, including neuropsychiatric and neurodegenerative conditions (*Leung et al., 2022*; *Chi, 2005*). Although significant progress has been made in understanding HD–DNA interactions,

uncovering the pathogenetic mechanisms of disease-causing missense mutations in HD have proven challenging.

The retina has long been used as a model system to study the role of HD TFs in normal central nervous system development and in neurological diseases (*Zagozewski et al., 2014*). During retinogenesis, HD TFs play essential roles in the patterning of neuroepithelium, specification of retinal progenitors and differentiation of all retinal cell classes that derive from a common progenitor (*Diacou et al., 2022*). Importantly, many HD TFs are shared between the brain and the retina during development and mutations in these TFs can lead to disease manifestation in both tissues (*Beby and Lamonerie, 2013*; *Henderson et al., 2009*; *Abouzeid et al., 2009*; *Lima Cunha et al., 2019*; *Voronina et al., 2004*; *Abouzeid et al., 2012*). The accessibility and wealth of available molecular tools make the retina a valuable tool to decipher the pathogenic mechanisms of HD TF mutations associated with neurological diseases.

Here, we study CRX, a HD TF essential for photoreceptor cells in the retina, as a model to understand how single amino acid substitutions in the HD impact TF functions and cause blinding diseases. Photoreceptors are the most numerous neurons in the retina and are specialized to sense light and initiate vision through a process called phototransduction. Animal studies have demonstrated that *Crx* is first expressed in post-mitotic photoreceptor precursors (*Muranishi et al., 2011*) and maintained throughout life (*Chen et al., 1997*; *Furukawa et al., 1997*). Loss of CRX results in impaired photoreceptor gene expression, failure of maturation and rapid degeneration of immature, non-functional photoreceptors (*Furukawa et al., 1999*). Protein-coding sequence variants in human *CRX* have been associated with inherited retinal diseases (IRDs) that affect photoreceptors: Leber congenital amaurosis (LCA), cone–rod dystrophy (CoRD), and retinitis pigmentosa (RP) (OMIM:602225). However, the recessive phenotype observed in *Crx* knockout mouse models fails to recapitulate many dominant human *CRX* mutations that arise de novo (*Furukawa et al., 1999*).

CRX contains two functional domains – the N-terminal HD and C-terminal activation domain (AD) (*Figure 1A*), both are required for proper activation of target genes and maintenance of normal *Crx* mRNA transcript abundance (*Chau et al., 2000*; *Chen et al., 2002*). To understand how CRX HD mutations cause diseases, we have previously reported a mutation knock-in mouse model carrying a hypomorphic mutation p.R90W (R90W) in CRX HD (*Tran et al., 2014*; *Ruzycki et al., 2015*). We found that R90W mutation produces a recessive loss-of-function phenotype very similar to that of *Crx* knockout mice.

Intriguingly, several missense mutations within the same HD recognition helix as R90W, including p.E80A (E80A) and p.K88N (K88N), are linked to severe dominant IRDs (*Freund et al., 1997*; *Swaroop et al., 1999*; *Nichols et al., 2010*; *Figure 1A, B*). Here, we utilized a multi-omics approach to investigate the functional consequences of the E80A and K88N mutations on CRX regulatory activities and photoreceptor development (*Figure 1—figure supplement 1*). Comparison of the in vitro HD DNA-binding models of CRX and disease variants generated by Spec-seq revealed unique specificity changes of each mutant protein. Introduction of each mutation into the endogenous *Crx* locus generated knock-in mouse models *Crx^{E80A}* and *Crx^{K88N}* that reproduced dCoRD- and dLCA-like phenotypes. ChIP-seq analysis of CRX-binding in vivo revealed mutation-specific changes in CRX targetome, consistent with mutation-specific DNA-binding changes in vitro. Retinal RNA-seq analysis uncovered two distinct mechanisms by which the two HD missense mutations contribute to altered gene expression programs during photoreceptor differentiation and maturation. Our results highlight the importance of residues E80 and K88 in CRX-mediated transcriptional regulation during photoreceptor development and the diverse mechanisms by which HD missense mutations can affect TF functions and lead to severe dominant neurological diseases.

## Results

### K88N but not E80A mutation alters CRX HD DNA-binding specificity in vitro

CRX belongs to the paired-like HD TF family that recognize a 6-bp DNA motif in a stereotypic way (*Desplan et al., 1988*; *Trelsman et al., 1989*; *Hanes and Brent, 1989*; *Hanes and Brent, 1991*; *Ades and Sauer, 1995*; *Wilson et al., 1995*; *Gehring et al., 1994*). Extensive studies of the HD have revealed a canonical HD–DNA recognition model where recognition of the 3' region (bases 4–6) of

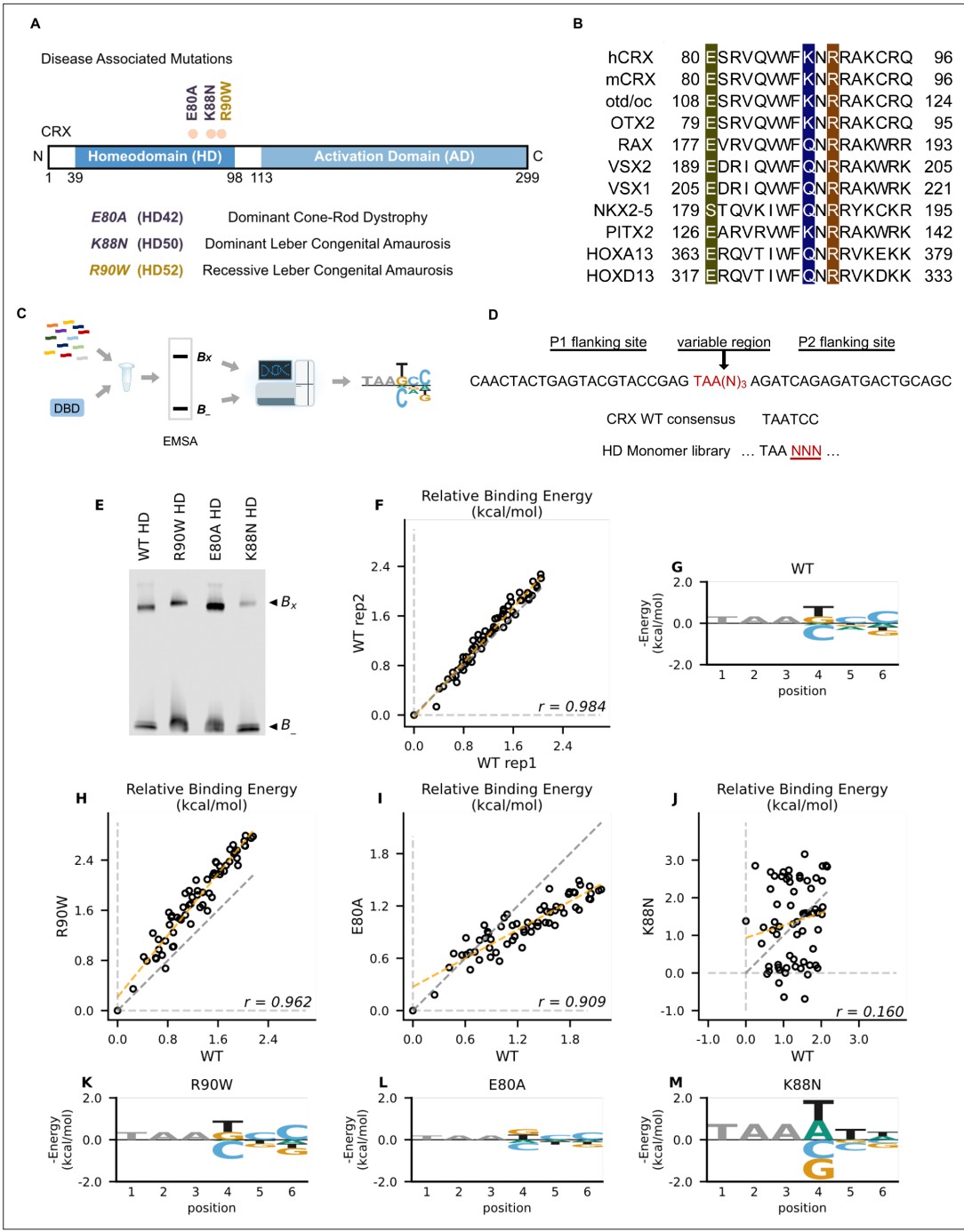

**Figure 1.** Disease associated missense mutations altered CRX homeodomain (HD) DNA-binding specificity (legend on the next page). (**A**) Diagram of CRX functional domains: HD for DNA-binding and activation domain (AD) for target gene transactivation. The three missense mutations in this study are located at the C-terminus of CRX HD and associated with different retinal diseases in human. Number in the parenthesis denotes the CRX HD position of the corresponding mutated residue. (**B**) Alignments of HD recognition helix sequences for the indicated HD proteins for which HD missense mutations have been associated with inherited diseases. Accession numbers can be found in *Supplementary file 1a*. Missense variants in this study (highlighted) are located at highly conserved residues across species and different HD transcription factors (TFs). (**C**) Spec-seq experimental workflow (Methods). (**D**) Spec-seq library design of monomeric HD-binding sites. (**E**) Electrophoretic mobility shift assay (EMSA) gel images of Spec-seq experiments with different CRX HD species. B$_x$: bound. B.: unbound. (**F**) Relative binding energy comparison from two different experiments with wild-type (WT) HD. (**G**) Binding energy model for WT CRX HD. Relative binding energy comparison between WT HD and R90W HD (**H**), E80A HD (**I**), or K88N

*Figure 1 continued on next page*

*Figure 1 continued*

HD (**J**). Consensus sequence is defined to have relative binding energy of 0kT (TAATCC for WT, R90W and E80A, TAATTA for K88N). The identity line is represented in gray dash. The orange dashed line shows the best linear fit to the data. Binding energy models for R90W HD (**K**), E80A HD (**L**), and K88N HD (**M**). Only sequence variants within two mismatches to the corresponding consensus sequences were used to generate binding models. Negative binding energy is plotted such that bases above the *x*-axis are preferred bases and bases below the *x*-axis are unfavorable bases. Constant bases (TAA) carried no information are drawn at arbitrary height in gray.

The online version of this article includes the following source data and figure supplement(s) for figure 1:

**Source data 1.**

**Source data 2.**

**Figure supplement 1.** Multi-omics approach to investigate the functional consequences of dominant disease mutations on CRX regulatory activities and photoreceptor development.

**Figure supplement 2.** Reversed-strand Spec-seq library showed similar changes in mutant CRX homeodomain (HD) DNA-binding specificity.

**Figure supplement 2—source data 1.**

---

the HD DNA-binding site is mediated by specificity determinants within the conserved HD recognition helix, corresponding to CRX residues 80–96 (*Desplan et al., 1988*; *Trelsman et al., 1989*; *Hanes and Brent, 1989*; *Hanes and Brent, 1991*; *Gehring et al., 1994*; *Noyes et al., 2008*; *Figure 1B*). In particular, HD residue 50, equivalent to CRX K88 residue (*Figure 1A*), is the major specificity determinant in paired-like HD TF–DNA interactions (*Trelsman et al., 1989*; *Hanes and Brent, 1989*). Since the three disease-associated HD missense mutations, E80A, K88N, and R90W, are located within the CRX HD recognition helix, we wondered if these mutations change CRX HD DNA-binding specificity.

We adapted a high-throughput in vitro assay, Spec-seq, that determines protein–DNA-binding specificity by sequencing (*Stormo et al., 2015*; *Zuo et al., 2017*; *Zuo and Stormo, 2014*). Spec-seq was developed based on the traditional electrophoretic mobility shift assay (EMSA) to measure protein–DNA interactions. Spec-seq allows us to measure the relative binding affinities (i.e., specificity) for a library of HD-binding motifs in parallel and generate quantitative binding models for different CRX HDs (*Figure 1C*). Based on the HD–DNA interaction model, we designed and tested a Spec-seq library containing all possible monomeric HD motifs (TAANNN) (*Figure 1D*).

We first obtained the wild-type (WT) CRX HD DNA-binding model with Spec-seq using bacterially expressed and affinity-purified HD peptides (*Figure 1E–G*, Methods). Relative binding energies of CRX WT HD from two experiments showed strong correlation ($r$: 0.984) and noise level (0.114 kT) within the expected range in typical Spec-seq data (*Figure 1F*). Binding energy model of WT HD was then generated by applying multiple linear regressions on the relative binding energies of all sequences within two basepair mismatches to the WT CRX consensus (TAATCC) (*Chen et al., 1997*; *Figure 1G*, Methods). A clear preference for CC bases at the 3′ end of the motif is consistent with known CRX-binding preference in vitro and in vivo (*Corbo et al., 2010*; *Kwasnieski et al., 2012*).

We next sought to understand how disease mutations affect CRX DNA-binding specificity. We purified all mutant HD peptides following the same protocol as WT HD peptides and verified their DNA binding (*Figure 1—figure supplement 2A-D*; *Chen et al., 2002*). Comparison of the relative binding energies between each pair of mutant and WT HD revealed distinct effects (*Figure 1H–J*). By definition, the consensus DNA-binding motif of a testing peptide has a relative binding energy of 0 kT. The relative binding energy difference between nucleotide variants and the consensus motif correlates with the DNA-binding specificity of the testing peptide. We found that R90W HD and E80A HD both prefer the same consensus motif as WT (*Figure 1K, L*). R90W HD bound with slightly higher specificity than WT, as demonstrated by most data points falling above the identity line (*Figure 1H, K*), suggesting that R90W HD is more sensitive than WT to binding sequence variations. In contrast, when comparing E80A HD with WT HD, many data points fell below the identity line and the relative binding energies regressed toward 0 on the E80A axis (*Figure 1I, L*). This suggests that E80A HD bound with lower specificity than WT HD and thus was more tolerant to base variations in the HD DNA motif. Different from R90W and E80A, K88N mutation dramatically altered CRX HD DNA-binding specificity ($r$: 0.160) (*Figure 1J, M*). The K88N preferred binding sequence (TAAT/ATT/A) is referred to as N88 HD motif hereafter. K88N HD also had the largest degree of discrimination from its preferred

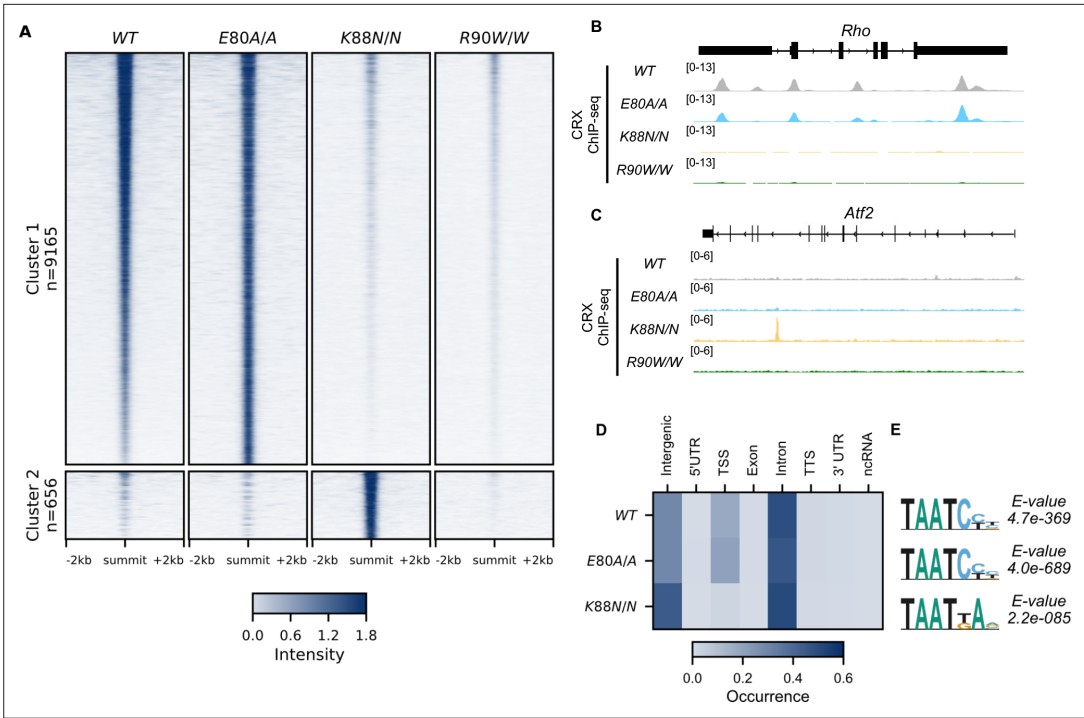

**Figure 2.** CRX E80A binds to wild-type (WT) sites while CRX K88N occupies novel genomic regions enriched for N88 homeodomain (HD) motif in vivo. (**A**) Enrichment heatmap depicting CRX ChIP-seq normalized reads centered at all possible CRX peaks ±2 kb, sorted by binding intensity in *WT* samples. Clusters were defined by hierarchical clustering of CRX-binding intensity matrix from all genotypes (Materials and methods). (**B, C**) Genome browser representations of ChIP-seq normalized reads for different CRX species in P14 WT and mutant mouse retinas at *Rho* and *Atf2*. (**D**) Enrichment heatmap showing fraction of CRX ChIP-seq peaks fall in different genomic environments. (**E**) Logo representations of de novo found short HD motifs under CRX ChIP-seq peaks in WT and mutant mouse retinas with DREME *E*-value on the right.

The online version of this article includes the following source data and figure supplement(s) for figure 2:

**Figure supplement 1.** Wild-type (WT) and mutation knock-in mouse CRX sequences and genotyping identifications.

**Figure supplement 1—source data 1.**

---

to the weakest binding motif, suggesting that it is most sensitive to variants in the HD DNA motif. As a control, we tested a second library with the TAANNN sites on the reverse strand and obtained similar results (*Figure 1—figure supplement 2E–L*). Together, these results indicate that while E80A mutation does not affect CRX HD DNA-binding specificity, the K88N mutation dramatically alters the specificity in vitro.

## E80A protein binds to WT sites while K88N occupies novel genomic regions with N88 HD motifs in vivo

Next, we asked if changes in DNA-binding specificity affected mutant CRX chromatin binding in developing photoreceptors. We first created two human mutation knock-in mouse models, $Crx^{E80A}$ and $Crx^{K88N}$, each carrying a single base substitution at the endogenous *Crx* locus (*Figure 2—figure supplement 1A, B*, Methods. For concision, we use $Crx^{E80A}$ and $Crx^{K88N}$ when both heterozygous and homozygous mutants are being discussed). We confirmed that *Crx* mRNA was expressed at comparable levels in WT and mutant retinas (*Figure 2—figure supplement 1C*), and the full-length CRX proteins were readily detectable in the nuclear extracts from all samples (*Figure 2—figure supplement 1D*). We then obtained genome-wide binding profiles for each CRX variant by chromatin immunoprecipitation followed by sequencing (ChIP-seq) on mouse retinas at P14, a time when all retinal cell types are born, photoreceptor specification is completed in WT animals, and prior to any observed cell death in other CRX mutants previously characterized (*Tran et al., 2014*; *Bassett and Wallace,*

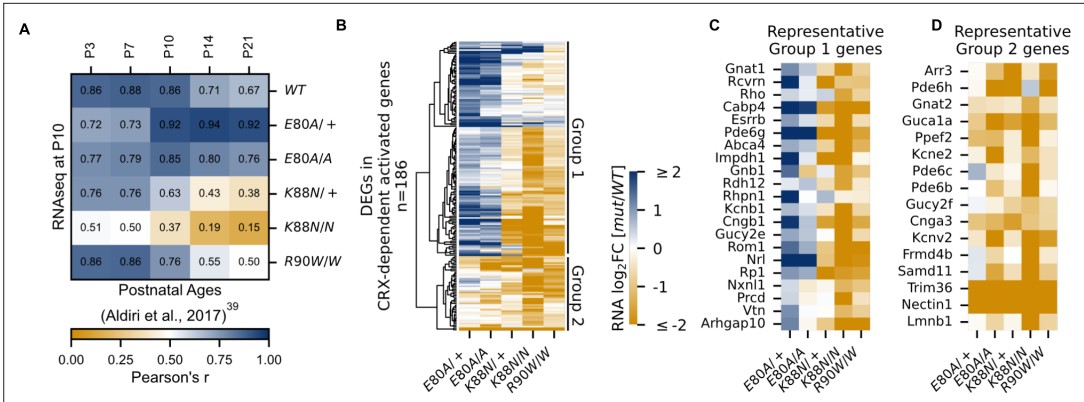

**Figure 3.** CRX-dependent activated genes affected in opposite directions in developing *Crx^E80A* and *Crx^K88N* mutant retinas. (**A**) Heatmap showing sample-wise Pearson correlations of the expression of all CRX-dependent activated genes between P10 wild-type (WT) and homeodomain (HD) mutant mouse retinas in this study (rows) with postnatal WT retinas from age P3 to P21 (columns, data from GSE87064). (**B**) Heatmap showing the expression changes of DEGs in CRX-dependent activated gene set in HD mutant mouse retinas at P10. (**C, D**) Heatmaps showing expression changes of selected photoreceptor genes from Groups 1 and 2. Color scale identical to (**B**).

The online version of this article includes the following figure supplement(s) for figure 3:

**Figure supplement 1.** Definition of CRX-dependent activated and CRX-independent gene sets.

**Figure supplement 2.** E80A and K88N mutation each causes novel gene expression changes in the CRX-independent category.

*2012*). To focus on changes specific to each mutant CRX protein, only homozygous animals were used for ChIP-seq profiling.

Unsupervised clustering of all CRX-binding sites revealed two major clusters (*Figure 2A*, Methods). Cluster 1 consisted of canonical WT CRX-binding sites that are also occupied by CRX E80A protein (*Figure 2A, B*). Similar to WT CRX, CRX E80A-binding in vivo was mostly enriched in intronic, followed by intergenic and transcription start site (TSS) regions (*Figure 3D*). In contrast, CRX R90W, that also showed similar consensus preference to WT in vitro, failed to produce significant DNA binding in vivo (*Figure 2A–C*, Methods). This suggests that the retinopathy phenotype of *Crx^R90W/W* is likely due to loss of binding at canonical WT CRX-binding sites. Intriguingly, while CRX K88N showed loss of binding at canonical CRX-binding sites, it gained a small set of binding sites (Cluster 2, *Figure 2A–C*). De novo motif searching with DREME (*Bailey, 2011*) under CRX peaks in each genotype revealed enrichment of monomeric HD motifs (*Figure 2E*) consistent with those found in Spec-seq (*Figure 1G, K–M*), highlighted by a change in enriched HD motif from WT CRX HD type to N88 HD type in the *Crx^K88N/N* retinas. Consistency with in vitro binding models suggests that in vivo changes in CRX chromatin binding were at least in part driven by the intrinsic changes in HD DNA-binding specificity by each individual mutation.

## E80A and K88N mutations affected the expression of CRX-dependent activated genes in opposite directions in a critical time window of photoreceptor differentiation

To understand how different CRX mutations affected gene expression at canonical and de novo binding sites and how these changes impair photoreceptor differentiation, we turned to bulk RNA-seq analysis from the developing retinas at P10. At P10, photoreceptors have started to differentiate, and the expression of many photoreceptor genes undergo exponential increase (*Aldiri et al., 2017*; *Kim et al., 2016*). To focus on the most relevant expression changes, we first defined a set of genes that most likely depend on CRX activity nearby for expression (*Figure 3—figure supplement 1A*, Methods). Briefly, we associated each CRX ChIP-seq peak to the nearest gene, filtered only genes with nearby CRX ChIP-seq peaks, and further narrowed the list of genes to those significantly down-regulated in the loss-of-function mutant *Crx^R90W/W*. Gene ontology (GO) analysis confirmed that this putative CRX-dependent gene set is associated with biological processes related to photoreceptor

development and functions (*Figure 3—figure supplement 1B*). This set of putative CRX-dependent genes also showed developmental dependent gain in expression, consistent with CRX's primary function as a transcriptional activator (*Figure 3—figure supplement 1C*). As a control, CRX-independent genes were constitutively expressed and largely involved in general cellular processes (*Figure 3—figure supplement 1D–F*). Therefore, the CRX-dependent gene set comprises genes important for photoreceptor differentiation and functional maturation and are dependent on CRX for activation. We denote these genes as 'CRX-dependent activated genes' (CRX-DAGs).

Next, we sought to understand how each mutation affected photoreceptor differentiation. One way of measuring the progression of photoreceptor differentiation is to determine the similarity in CRX-DAG expression in a given sample with that of known developmental ages in WT control animals. We thus performed sample-wise correlation of *CRX-DAG* expression obtained in our P10 samples with a previously published RNA-seq dataset of normal mouse retinal development (*Figure 3A*; *Aldiri et al., 2017*). As expected, our P10 WT sample showed strong correlations with all developmental ages in the published WT control dataset. A stronger correlation with early ages (P3, P7, P10) and a weaker correlation with later ages (P14, P21) is also an indication of ongoing photoreceptor differentiation at P10. Unlike the WT sample, $Crx^{E80A/+}$ and $Crx^{E80A/A}$ samples both showed a stronger correlation with later developmental ages (P14, P21) but a weaker correlation with earlier postnatal ages (P3, P7). Since the *CRX-DAGs* are normally developmentally upregulated, this shift in correlation toward later developmental ages suggested that these genes were prematurely upregulated in the P10 $Crx^{E80A/+}$ and $Crx^{E80A/A}$ mutant retinas. In contrast, $Crx^{K88N/+}$ and $Crx^{K88N/N}$ samples both showed a weaker correlation with all developmental ages when compared with WT samples in our dataset. This suggests that early photoreceptor differentiation was compromised in both $Crx^{K88N}$ mutants, consistent with their association with early-onset LCA (*Nichols et al., 2010*). Importantly, $Crx^{R90W/W}$, also associated with LCA-like phenotype (*Tran et al., 2014*; *Swaroop et al., 1999*), displayed strong correlation with earlier ages (P3, P7) similar to WT, but only showed moderate correlation with later ages (P14, P21). This suggests loss of CRX function at canonical binding sites does not affect the initiation of photoreceptor differentiation, but WT CRX activity at these sites is required to sustain differentiation. Since *CRX-DAG* expression was more severely affected in $Crx^{K88N}$ mutants than in $Crx^{R90W/W}$, the photoreceptor differentiation deficits seen in the $Crx^{K88N}$ mutants cannot be explained solely by the loss of regulatory activity at canonical CRX-binding sites. Overall, our sample-wise correlation analysis with normal retinal development dataset suggests that E80A and K88N mutations affected the expression CRX-dependent activated genes in opposite directions, implicating novel and distinct pathogenic mechanisms from the loss-of-function R90W mutation.

## $Crx^{E80A}$ retinas show upregulation of rod genes but downregulation of cone genes, underlying CoRD-like phenotype

Upon closer examination, we noted that not all *CRX-DAGs* were upregulated in $Crx^{E80A}$ mutants (*Figure 3B*). Hierarchical clustering of all *CRX-DAGs* using expression changes revealed two major groups (Methods). In aggregate, when compared to WT, Group 1 genes were upregulated in $Crx^{E80A}$ mutants while Group 2 genes were downregulated. We noted genes indicative of the two photoreceptor subtypes, rods and cones, could partially define the two groups (*Figure 3C, D*). For example, *Esrrb* (*Onishi et al., 2010*) and *Nrl* (*Yoshida et al., 2004*) in Group 1 are important regulators of rod differentiation. Other genes in Group 1 are components of the phototransduction cascade in rods, including *Rcvrn* (*Zang and Neuhauss, 2018*), *Rho* (*Palczewski, 2006*), *Gnat1* (*Dryja et al., 1996*; *Carrigan et al., 2016*), *Pde6g* (*Dvir et al., 2010*), *Abca4* (*Allikmets et al., 1997*; *Nasonkin et al., 1998*), *Gnb1* (*Kitamura et al., 2006*), *Rdh12* (*Janecke et al., 2004*), *Cngb1* (*Bareil et al., 2001*), and *Rp1* (*Pierce et al., 1999*; *Sullivan et al., 1999*). Mis-regulations of many of these genes have been associated with diseases that affect rod development, function, and long-term survival. The increased activation of these genes likely underlies the stronger correlation with later developmental ages in $Crx^{E80A}$ mutant retinas (*Figure 3A*). In contrast, Group 2 genes, many downregulated in $Crx^{E80A}$ mutants, were implicated in cone development and functions. For example, *Gnat2* (*Kohl et al., 2002*; *Rosenberg et al., 2004*), *Pde6c* (*Thiadens et al., 2009*), and *Pde6h* (*Kohl et al., 2012*; *Piri et al., 2005*) all act in the cone phototransduction cascade. Mis-regulation of these genes has also been implicated in different retinal dystrophies that primarily affect cone photoreceptors. Comparison of ChIP-seq signal revealed that peaks associated with

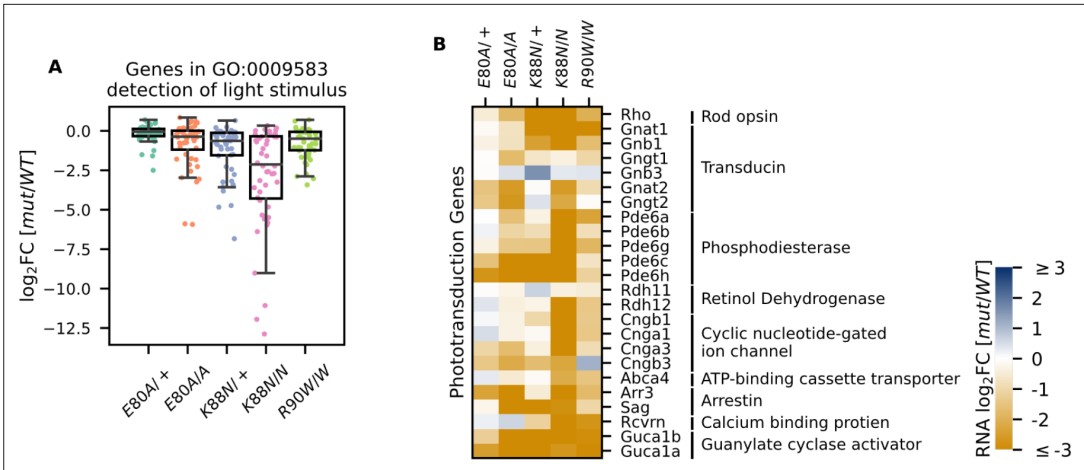

**Figure 4.** Photoreceptor genes important for phototransduction are downregulated in all homeodomain (HD) mutants. (**A**) Box plot showing that genes in the detection of light stimulus gene ontology (GO) term were downregulated and affected to various degrees in different adult (P21) HD mutant mouse retinas. (**B**) Heatmap showing that expression of both cone and rod phototransduction genes were downregulated in adult (P21) HD mutant mouse retinas. Annotation of rod and cone enrichment of each gene is in *Supplementary file 1f*. See *Figure 4—figure supplement 1* for the developmental expression dynamics of these genes.

The online version of this article includes the following figure supplement(s) for figure 4:

**Figure supplement 1.** Developmental expression pattern of phototransduction genes in *WT* animals.

Group 2 genes showed lower occupancy compared to Group 1 genes in the *Crx^{E80A/A}* retinas (Mann–Whitney *U*-test p-value: 9.51e−07) while no difference was observed in WT retinas (Mann–Whitney *U*-test p-value: 0.541), suggesting loss of CRX activity likely underlies the downregulation of Group 2 genes in the *Crx^{E80A}* mutants (*Figure 3—figure supplement 2A*). Collectively, the selective down-regulation of cone genes in Group 2 may explain the CoRD-like phenotype in adult *Crx^{E80A}* mutant mice described later.

Additionally, we noticed that a subset of genes not affected in *Crx^{R90W/W}* (CRX-independent genes) were also downregulated in *Crx^{E80A}* mutants (*Figure 3—figure supplement 2B–D*). Among these genes were transcription regulators important for early photoreceptor development, such as *Ascl1* (*Kaufman et al., 2019*), *Rax* (*Irie et al., 2015*), *Sall3* (*de Melo et al., 2011*), and *Pias3* (*Onishi et al., 2009*). The downregulation of these factors coincided with the upregulation of mature rod genes in P10 *Crx^{E80A}* retinas, suggesting that the E80A mutation might hamper the proper timing of photoreceptor differentiation.

## *Crx^{K88N}* retinas display greater reduction of rod and cone genes than the loss-of-function mutants

*Crx^{K88N}* retinas had the most severe gene expression changes among all mutants with downregu-lation of both Group 1 (rod) and Group 2 (cone) genes (*Figure 3B–D*). The heterozygous *Crx^{K88N/+}* retina displayed a similar degree of expression reduction as homozygous *Crx^{R90W/W}*, consistent with its association with dLCA (*Nichols et al., 2010*). Given the normal phenotype of heterozygous loss-of-function mutants – *Crx^{+/−}* and *Crx^{R90W/+}*, these results suggest that mutant CRX K88N not only failed to activate WT target genes, but also functionally antagonized WT CRX regulatory activity in differentiating photoreceptors (*Tran et al., 2014*). This antagonism might be associ-ated with ectopic CRX K88N activity when bound to regulatory regions with N88 HD DNA motifs (+/−). In the absence of WT CRX, *Crx^{K88N/N}* retina displayed a more severe expression reduction of *CRX-DAG* than *Crx^{R90W/W}*, raising the possibility that CRX K88N also antagonized the activity of other transcriptional regulators important for photoreceptor differentiation. Supporting this possi-bility, a set of CRX-independent genes were also mis-regulated in both heterozygous and homo-zygous *Crx^{K88N}* mutants. We noted that a number of these downregulated genes are also involved in photoreceptor functional development (*Figure 3—figure supplement 2B, E, F*). Overall, CRX

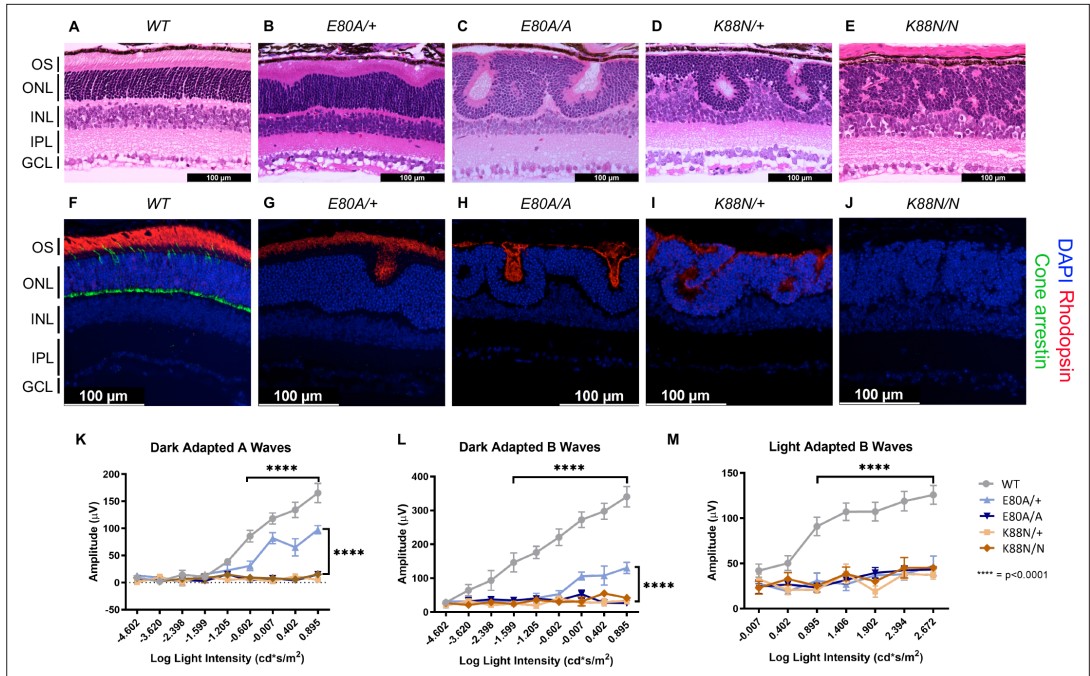

**Figure 5.** Only *Crx^E80A/+* retinas maintain photoreceptor OS and residual rod electroretinogram (ERG) response. (**A–E**) Hematoxylin–eosin (H&E) staining of P21 retina sections show that photoreceptor OS layer is absent in all mutant retinas except *Crx^E80A/+*. OS: outer segment; ONL: outer nuclear layer; INL: inner nuclear layer; IPL: inner plexiform layer; GCL: ganglion cell layer. Scale bar, 100 µm. (**F–J**) Rhodopsin (RHO, red) immunostaining is present in *Crx^E80A/+*, *Crx^E80A/A*, and *Crx^K88N/+* retinas and absent in *Crx^K88N/N* retina. Cone arrestin (mCAR, green) immunostaining is absent in all mutant retinas. Nuclei were visualized by (4',6-diamidino-2-phenylindole) DAPI staining (blue). Scale bar, 100 µm. (**K–M**) The ERG responses recorded from 1 month mice. Error bars represent the standard error of the mean (SEM, $n \geq 4$). p-value: Two-way analysis of variance (ANOVA) and Tukey's multiple comparisons. ****$p \leq 0.0001$. ns: $>0.05$.

K88N is associated with greater gene expression changes than other CRX HD mutants, which is likely attributed to ectopic regulatory activity.

## Both *E80A* and *K88N* mutants show compromised rod/cone terminal differentiation in young adults

Since *Crx^+/−* and *Crx^R90W/+* mutant mouse models showed a late-time recovery in photoreceptor gene expression and function (*Ruzycki et al., 2015*), we sought to determine the degree of photoreceptor differentiation in *Crx^E80A* and *Crx^K88N* mutants at P21. At this age, the normal retina has largely completed terminal differentiation with photoreceptor gene expression reaching a plateau (*Aavani et al., 2017*). However, when P21 CRX HD mutant retinas were examined for the expression of genes under the GO term detection of light stimulus (GO:0009583), which comprises genes in both rod and cone phototransduction cascades, many genes failed to reach WT levels, despite variable degrees of impact across different HD mutants (*Figure 4A*). *Crx^E80A* mutants, in contrast to the increased rod gene expression at P10, displayed a deficiency in both cone and rod phototransduction genes at P21 (*Figure 4B*, *Supplementary file 1h*). This suggests that mutant CRX E80A transcriptional activity fails to sustain photoreceptor terminal differentiation and ultimately results in non-functional and severely affected photoreceptors. In comparison, *Crx^K88N* mutants showed severely reduced expression of rod/cone phototransduction genes at both P10 and P21 (*Figures 3B–D and 4B*).

To assess retinal morphology and photoreceptor subtype-specific gene expression at the cellular level, we performed immunohistochemistry analysis on P21 retinal sections. In WT animals, a hallmark of photoreceptor maturation is the outgrowth of photoreceptor outer segments (OS) filled with proteins necessary for the phototransduction. We thus performed hematoxylin and eosin (H&E) staining on P21 sagittal retinal sections to visualize changes in retinal layer organization, focusing on photoreceptor layers – outer nuclear layer (ONL) and OS. Compared to the well-organized ONL in

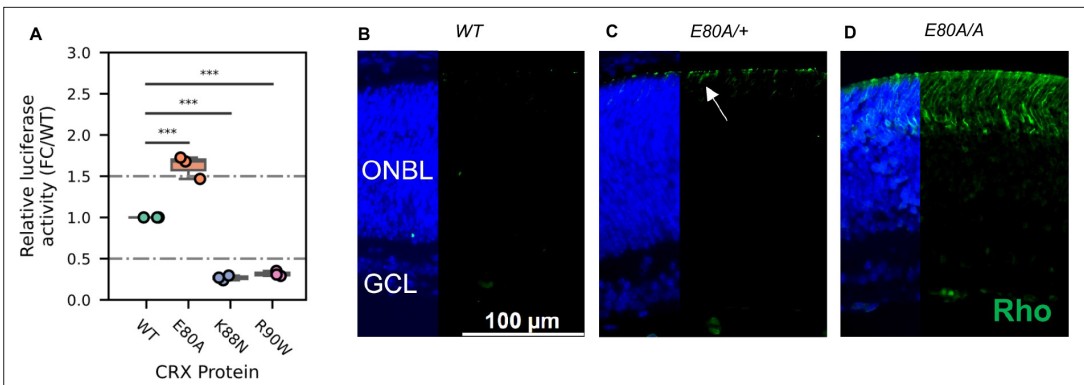

**Figure 6.** CRX E80A hyperactivity underlies precocious photoreceptor differentiation in $Crx^{E80A}$ retinas. (**A**) Boxplot showing luciferase reporter activities of different CRX variants. p-values for one-way analysis of variance (ANOVA) with Turkey honestly significant difference (HSD) test are indicated. (**B–D**) Rhodopsin (RHO, green) immunostaining is absent in P3 wild-type (WT) retina but detected in $Crx^{E80A/+}$ and $Crx^{E80A/A}$ retinas. Nuclei are visualized by DAPI staining (blue). Arrow indicates the sporadic RHO staining in $Crx^{E80A/+}$ sample. ONBL: outer neuroblast layer; GCL: ganglion cell layer. Scale bar, 100 µm.

The online version of this article includes the following figure supplement(s) for figure 6:

**Figure supplement 1.** Cone photoreceptors born in $Crx^{E80A}$ retinas and hyperactivity of CRX E80A at S-opsin promoter.

WT retinas (*Figure 5A*), all mutants showed variable degrees of ONL disorganization, forming waves, whorls, and rosettes (*Figure 5B–E*). The ONL disorganization was more severe in homozygotes than in heterozygotes for both mutations and $Crx^{K88N}$ mutants were more severely affected than $Crx^{E80A}$ mutants. Photoreceptor OS layer was formed in the $Crx^{E80A/+}$ retinas, but absent in $Crx^{E80A/A}$, $Crx^{K88N/+}$, and $Crx^{K88N/N}$ mutant retinas. Inner retinal layers, including inner plexiform layer (IPL) and ganglion cell layer were not as severely affected as the outer retinal layers, supporting a model that the mutant morphological abnormalities largely originated from the diseased photoreceptors. These morphological abnormalities were distinct from the degenerative phenotypes of other *Crx* mutant models reported previously (*Tran et al., 2014*; *Tran and Chen, 2014*; *Roger et al., 2014*).

Next, we sought to determine the expression of the rod-specific visual pigment rhodopsin (RHO) and cone arrestin (mCAR) in the P21 mouse retinas. In WT retinas, RHO is trafficked to the rod OS while mCAR is present in the cone OS and IS (inner segment), cell body, and synaptic terminals (*Figure 5F*). Unlike WT retina, all mutants lacked mCAR immunoreactivity (*Figure 5G–J*), consistent with the loss of cone gene expression shown by RNAseq (*Figure 4B*). In $Crx^{E80A/+}$ retinas, RHO staining was localized to the OS layer; in $Crx^{E80A/A}$ and $Crx^{K88N/+}$ retinas, positive RHO staining was observed within the whorls and rosettes; in $Crx^{K88N/N}$ mutant retinas, RHO staining was completely absent. Importantly, we did not observe mis-localized RHO staining in the inner retinal layers (INL) suggesting that the developmental programs of other retinal cell types were not directly affected by E80A or K88N mutation. Overall, abnormalities in the cone/rod gene expression matched the corresponding human disease diagnosis (*Freund et al., 1997*; *Nichols et al., 2010*), and the phenotypic severity correlated with the degree of mis-regulation of CRX target genes in the corresponding RNAseq dataset. Thus, these results support a model that CRX HD mutation-mediated mis-regulation of gene expression disrupts photoreceptor terminal differentiation and leads to defects in retinal layer organization and OS formation.

## *E80A* and *K88N* mouse models show visual function deficits that recapitulate human diseases

To understand the consequences of disrupted photoreceptor differentiation on visual function, we measured electroretinogram (ERG) responses to light stimuli for WT and mutant mice at 1 month of age (*Figure 5K–M*). In response to incremental changes of light intensities, WT animals showed corresponding amplitude increases in dark-adapted A-waves (rod signals) and B-waves (rod-evoked bipolar cell signals), as well as in light-adapted B-waves (cone-evoked bipolar cell signals). The three severe mutants, $Crx^{E80A/A}$, $Crx^{K88N/+}$, and $Crx^{K88N/N}$ had no detectable dark- or light-adapted ERG responses,

suggesting that these mice have no rod or cone function and are blind at young ages. The null ERG phenotype of the $Crx^{K88N}$ animals is consistent with the clinical LCA phenotype in humans (**Nichols et al., 2010**). In contrast, $Crx^{E80A/+}$ animals retained partial rod ERG responses as indicated by the reduced A- and B-wave amplitudes (**Figure 5K, L**). Yet, $Crx^{E80A/+}$ animals had no detectable cone ERG responses, which is consistent with the CoRD clinical phenotype in humans (**Freund et al., 1997**; **Figure 5M**). Taken together, the visual function impairment in each CRX HD mutant model, coincided with the morphological and molecular changes, suggesting that $Crx^{E80A}$ and $Crx^{K88N}$ mouse models recapitulate the corresponding human diseases.

## CRX E80A has increased transactivation activity and leads to precocious differentiation in $Crx^{E80A}$ retinas

Lastly, we asked what might be the molecular mechanism that causes the mis-regulation of photoreceptor genes in the mutant retinas. Previous studies have established that reporter assays with the *Rhodopsin* promoter in HEK293T cells can measure changes of CRX transactivation activity and inform the mechanisms by which photoreceptor genes are mis-regulated in CRX mutant retinas (**Chen et al., 2002**). We thus tested the transactivation activity of the three CRX HD mutants on the pRho-Luc reporter in HEK293T cells (**Figure 6A**, Methods). Consistent with published studies, R90W mutant had significantly reduced activity compared to WT CRX (**Chen et al., 2002**). K88N mutant showed a similarly reduced activity as R90W, consistent with K88N's loss of binding at canonical CRX sites and failure to activate *CRX-DAGs* in vivo (**Figures 2A, B and 3A–C**). In contrast, E80A mutant, which binds to canonical CRX sties, showed significantly increased transactivation activity on the *Rho* promoter. This hyperactivity of E80A protein at *Rho* promoter correlates with the upregulation of *CRX-DAGs* in the mutant retinas at P10 (**Figure 3A–C**).

A transition to the next developmental stage usually requires the expression of important developmental genes passing an abundance threshold. Based on the hyperactivity model, photoreceptor genes are activated stronger in $Crx^{E80A}$ retinas and thus could reach the abundance threshold earlier. To determine the consequences of E80A hyperactivity on photoreceptor differentiation timing, we compared RHO protein expression during early postnatal retinal development using retinal section immunostaining (**Figure 6B–D**). In WT retinas, most rods were born by P3 but had not differentiated (**Figure 6B**). Previous studies showed that RHO proteins were detected by IHC starting around P7 in WT retinas (**Aavani et al., 2017**). In comparison, both $Crx^{E80A/+}$ and $Crx^{E80A/A}$ retinas showed positive RHO staining at P3 (**Figure 6C, D**). RHO+ cells were largely seen in the outer portion of the ONBL layers in $Crx^{E80A/+}$ retinas, and strikingly spread throughout the large presumptive ONL layers in $Crx^{E80A/A}$ retinas. The detection of RHO protein in P3 $Crx^{E80A}$ mutant retinas indicates that photoreceptor differentiation program was precociously activated. Taken together, our results support a model that *E80A* and *K88N* mutations each perturbs CRX regulatory activity in a unique way, causes photoreceptor differentiation defects, and ultimately leads to distinct dominant disease phenotype that recapitulates human diseases.

## Discussion

Through molecular characterization of mutant proteins, transcriptome, and cellular profiling of developing mutant mouse retinas and ERG testing of adult retinas, we have identified two novel pathogenic mechanisms of CRX HD mutations, E80A and K88N, that are associated with dominant CoRD and dominant LCA in human (**Freund et al., 1997**; **Swaroop et al., 1999**). Distinct from the previously characterized loss-of-function R90W mutation (**Tran et al., 2014**; **Ruzycki et al., 2015**), E80A and K88N mutations produce altered CRX proteins with gain of regulatory functions – CRX E80A is associated with increased transcriptional activity and CRX K88N has altered DNA-binding specificity. Both CRX E80A and CRX K88N proteins impair photoreceptor gene expression, development and produce structural and functional deficits in knock-in mouse models, recapitulating human diseases (**Tran and Chen, 2014**; **Figure 7**). Thus, both target specificity and regulatory activity precision at the canonical CRX targets are essential for proper photoreceptor development and functional maturation.

Although associated with distinct disease phenotypes, the E80A, K88N, and R90W mutations are located very close to each other in the CRX HD recognition helix. Extensive biochemical and structural studies on the HD–DNA complexes afford important insights into how these mutations could affect

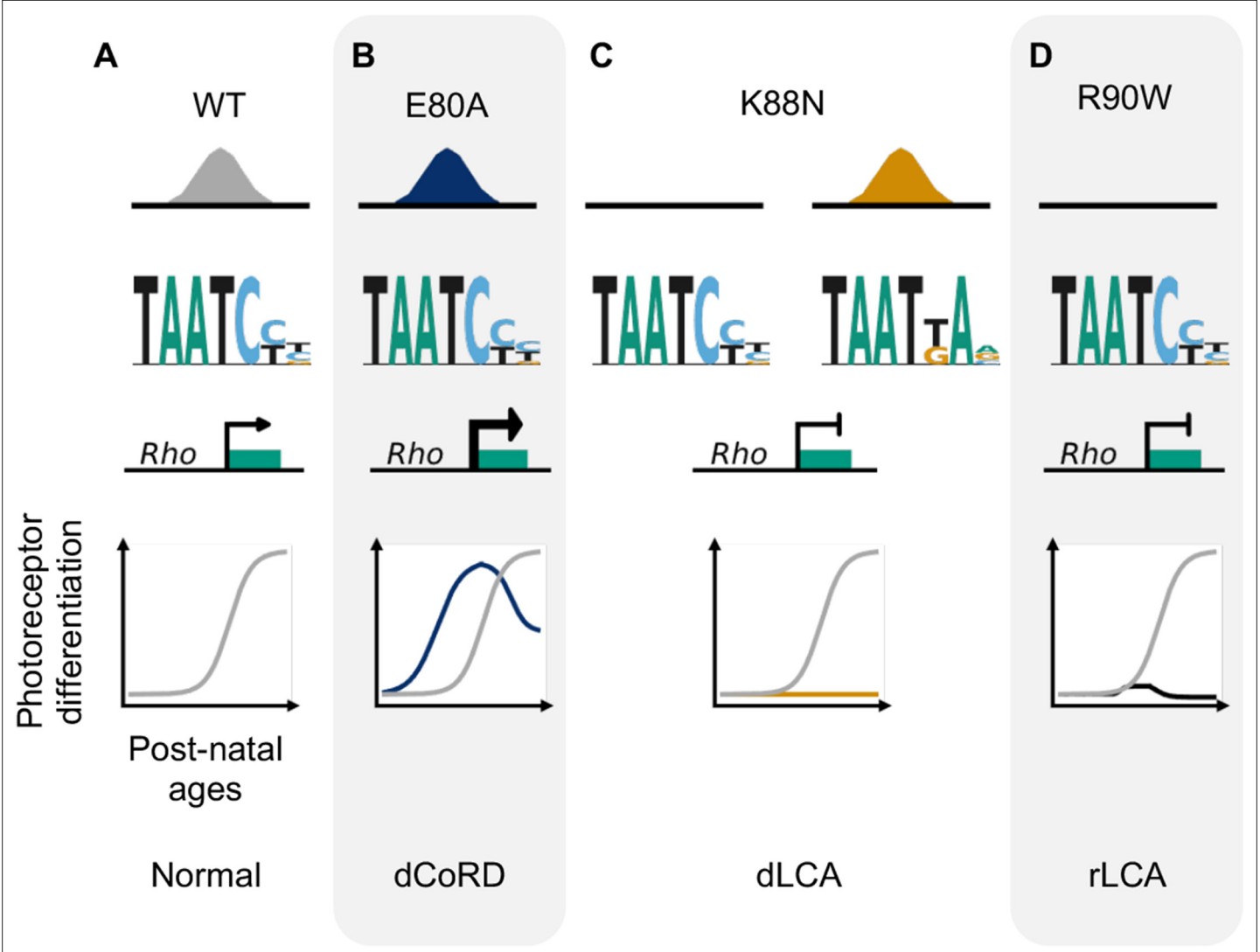

**Figure 7.** Missense mutations in CRX homeodomain (HD) affect photoreceptor gene expression and lead to distinct retinal disease phenotypes through gain- and loss-of-function mechanisms. dCoRD: dominant cone–rod dystrophy; dLCA/rLCA: dominant/recessive Leber congenital amaurosis.

CRX HD–DNA interactions differently. Most distinctively, CRX K88 residue, at HD50 position, is the major contributor to HD DNA-binding specificity, with lysine making favorable interactions with both guanines in the CRX consensus TAATCC-binding site (*Baird-Titus et al., 2006*; *Chaney et al., 2005*). It is thus expected that K88N mutation drastically changes CRX DNA-binding preference at the 3′ end of the HD motif, reminiscent of previous findings on novel HD DNA-binding specificity using bacterial one-hybrid (B1H) systems (*Noyes et al., 2008*; *Chu et al., 2012*). Supporting the importance of K88 residue-mediated CRX target specificity in regulating photoreceptor development, $Crx^{K88N}$ retinas show more severe perturbations in photoreceptor gene expression and development than in loss-of-function mutant $Crx^{R90W/W}$. $Crx^{K88N/+}$ and $Crx^{K88N/N}$ mice display the most severe photoreceptor morphological deficits observed in any $Crx$ mouse models and show absence of visual functions in young adults. Thus, CRX target specificity is critical for photoreceptor development fidelity.

In developing WT mouse retinas, the HD motif preferred by HD TFs with a glutamine (Q) at HD50 position encodes quantitatively different activity than the CRX consensus suggesting functional difference between HD-binding site variants (*Irie et al., 2015*; *Kimura et al., 2000*). It is likely that the severe $Crx^{K88N}$ phenotypes are attributed to both diminished activity at canonical CRX motifs and ectopic binding and transcriptional activity at N88 HD motifs. Since a functional copy of WT CRX is retained in $Crx^{K88N/+}$ retinas, the lack of WT activity alone cannot explain the severe developmental

deficits. Alternatively, these results suggest involvement of additional regulatory mechanisms: CRX K88N activity at N88 HD motifs might (1) ectopically activate genes whose expression prevents the progression of development or inactivate genes required for development; (2) interfere with other HD TFs that also recognize N88 HD motifs; (3) lead to epigenetic alterations that antagonize normal CRX functions. Many other HD containing TFs are expressed in developing mouse retina, including OTX2, RAX, VSX2, PAX6, SIX3/6, and LHX family (*Diacou et al., 2022*). Different from CRX, these HD TFs are essential for gene regulation in retinal progenitor cells and/or in other retinal cell lineages. Alteration of CRX DNA-binding specificity could mis-regulate genes originally targeted by these HD TFs and lead to severe perturbations in the retinal gene regulatory networks. To date, most studies have focused on CRX activity at cis-regulatory sequences enriched for the WT CRX consensus motifs (*Hughes et al., 2018*; *White et al., 2016*). Systematic comparison of regulatory activity at N88 HD motifs and WT consensus in the context of photoreceptor development in both WT and mutant retinas would be needed to substantiate the impact of mutant CRX K88N activity at different HD motifs. These experiments will also help clarify the pathogenic mechanisms in the *Crx^{K88N}* models and extend our knowledge of CRX HD-mediated regulatory grammar during photoreceptor development.

Different from CRX K88, the E80 and R90 residues, although the most common residues at HD42 and HD52 positions, respectively, do not contact DNA directly and thus lacked in-depth investigations in prior studies. CRX R90 residue has been suggested to confer additional stability for the HD fold besides the core residues and make contacts with the DNA backbone through bases in the TAAT core motif (*Baird-Titus et al., 2006*; *Chaney et al., 2005*). Substitution of the basic R90 residue with a bulky, neutral tryptophan (W) potentially reduces overall CRX HD stability which in term reduces CRX HD DNA-binding affinity without affecting its binding preference. The potential reduction in CRX HD–DNA complex stability is in line with our observation that R90W mutation abolishes CRX binding across the genome resulting in global loss of CRX target gene activation. It also explains the association of CRX R90W mutation with recessive loss-of-function LCA phenotypes in human and mouse (*Swaroop et al., 1999*; *Tran and Chen, 2014*).

Structural studies suggest that CRX E80 residue plays a role in stabilizing the HD–DNA-binding complex through intramolecular interactions with other HD residues (*Wilson et al., 1995*; *Chaney et al., 2005*), yet functional validations await further experiments. E80A mutation, replacing glutamic acid (E), which is acidic and polar, with alanine (A), which is neutral and non-polar, could render HD–DNA interactions more promiscuous as reflected in overall reduced magnitude of CRX E80A HD specificity (*Figure 1I*). Regulatory sequences of many photoreceptor genes contain both consensus and non-consensus CRX motifs (*Chen et al., 1997*; *Furukawa et al., 1997*; *Chen et al., 2002*; *Corbo et al., 2010*). It is likely that the more promiscuous CRX E80A–DNA interaction increases the likelihood of non-consensus CRX HD motifs being bound and activated resulting in overall increased transcriptional output (hyperactivity) as seen both in luciferase assays and in developing *Crx^{E80A}* mouse retinas. The promiscuous TF-DNA-binding associated hyperactivity phenomenon has also been observed in a dominant disease mouse model harboring a missense mutation in the zinc finger TF Krüppel-like factor-1 (KLF1) (*Gillinder et al., 2017*). Yet, in adult *Crx^{E80A}* retinas, photoreceptor terminal differentiation is impaired, resulting in disrupted retinal morphology and defective visual functions. Photoreceptor differentiation is programmed via sequential and concerted gene expression programs within a defined time window (*Wang and Cepko, 2016*; *Brzezinski and Reh, 2015*). One explanation for adult *Crx^{E80A}* phenotypes is that CRX E80A hyperactivity precociously activates later stage genes in the absence of proper nuclear context and/or subcellular structures, which in turn negatively impacts early events in photoreceptor differentiation. These observations underscore the importance of precisely tuned CRX-mediated transcriptional activity during photoreceptor development.

Although a global increase in expression is expected in *Crx^{E80A}* retinas based on the hyperactivity model, a subset of CRX-dependent activated genes implicated in cone photoreceptor development and functions is downregulated in both differentiating and mature mutant retinas. While cones undergo terminal differentiation to develop into cone subtypes – M- or S-cones – in a similar postnatal window as rods, they were born prenatally in mice within an earlier time window than rods. At an early postnatal age, cells expressing RXRγ, a ligand-dependent nuclear hormone receptor normally expressed in developing cones (*Kaufman et al., 2019*; *Sapkota et al., 2014*; *Mori et al., 2001*; *Roberts et al., 2005*), were observed in *Crx^{E80A}* retinas, suggesting cone photoreceptors were born in these mutant retinas (*Figure 6—figure supplement 1A*). One model for lack of cone markers in

adult *Crx*<sup>*E80A*</sup> retinas is that CRX E80A improperly activates later stage cone genes at a much earlier time window, disrupting cone terminal differentiation. Supporting this model, CRX E80A also hyper-activates the S-cone opsin promoter reporter pOpn1sw-luc (*Figure 6—figure supplement 1B*). An alternative model is that cones might be more sensitive to perturbations in CRX activity. It is known that cones depend on a different repertoire of TFs than rods for subtype terminal differentiation (*Kaufman et al., 2019*). It is possible that cone TFs respond differently to mutant CRX E80A hyperactivity, leading to the distinct expression changes in *Crx*<sup>*E80A*</sup> mutant retains. It is important to note that different point mutations at CRX E80 residue have been reported in dominant CoRD cases (ClinVar VCV000865803.1, VCV000007416.7, VCV000099599.6), emphasizing the importance of residue CRX E80 in regulating cone photoreceptor development. Since cones only make up a very small portion (3%) of photoreceptors in mouse retinas (*Carter-Dawson and LaVail, 1979*), quantitative character-ization of CRX E80A molecular functions in a cone dominant retina warrants further study to under-stand its selective effect on the cone differentiation program and help elucidate WT CRX regulatory principles in early photoreceptor development.

Given that the spatial structures and HD–DNA contact models of HD proteins are evolutionarily conserved, our study of CRX provides valuable molecular insights for HD mutations implicated in other diseases. For example, p.E79K substitution (corresponds to CRX E80) in OTX2 HD is associ-ated with dominant early-onset retinal dystrophy (*Vincent et al., 2014*), heterozygous p.R89G (corre-sponds to CRX R90) mutation in OTX2 HD causes severe ocular malformations (*Ragge et al., 2005*), and missense mutations of the CRX K88 and R90 homologous residues in PITX2 HD are associated with dominant Rieger syndrome (*Perveen, 2000*). It is likely that these mutations affect HD activity in similar ways as observed in CRX, and the exact disease manifestation is determined by cell-type- or tissue-specific mechanisms. The retina is readily accessible, and a broad range of molecular tools are available for ex vivo and in vivo manipulations. We believe that CRX is an ideal model to study the pathogenic mechanisms of HD mutations and to test therapeutic regimens, which would ultimately benefit the study of HD TFs and their associated diseases in other tissues and organs.

One limitation of this work is that effects of E80A and K88N mutations on CRX HD–DNA interac-tions have been evaluated at monomeric HD motifs and with homogenous protein species both in vitro and in vivo. Further evaluation of WT and mutant CRX binding at dimeric motifs will be desir-able, since selected dimeric HD motifs are known to mediate HD TF interactions to ensure gene expression fidelity during development (*Rister et al., 2015*; *Tucker and Wisdom, 1999*). Relatedly, we also need to address how CRX WT and mutant E80A or K88N proteins interact at HD-binding motifs – whether they cooperate or compete with each other, whether these interactions are HD motif sequence dependent, and how gene expression is impacted by CRX cooperativity or competition. While CRX HD mediates both TF–DNA interactions and protein–protein interactions, evaluation of how E80A and K88N mutations impact CRX interaction with other important photoreceptor TFs and how perturbations in these interactions lead to disease phenotypes warrant further study.

Collectively, our findings support a unifying model in which precise CRX interaction with cis-regulatory sequences is essential for gene expression and functional maturation during photoreceptor development. Disease-associated mutations in CRX have been classified into two main groups – inser-tion/deletion-derived frameshift mutations in the AD and missense mutations in the HD (*Tran and Chen, 2014*). Prior biochemical and mouse model studies of the first group have established that AD-truncated mutant proteins abolish CRX transcriptional activity and functionally interfere with the WT allele. As a result, the mutant retinas fail to activate or maintain robust cone/rod gene expression, resulting in incomplete photoreceptor differentiation and ultimately rapid degeneration of immature photoreceptors (*Furukawa et al., 1999*; *Tran et al., 2014*). In this study, we demonstrate that missense mutations in the CRX HD, by either a loss- or gain-of-function mechanism, alter CRX target specificity and/or CRX transactivation activity. These biochemical property changes impair CRX-mediated tran-scriptional regulation in vivo and lead to distinct morphological and functional deficits (*Figure 7A–D*). Despite the difference in molecular mechanisms, both *Crx*<sup>*E80A*</sup> and *Crx*<sup>*K88N*</sup> mouse models develop whorls and rosettes in the ONL by P21, which are not observed in degenerative CRX mouse models (*Tran et al., 2014*), suggesting distinct pathogenic mechanisms. Future cellular biology studies are needed to understand the formation mechanisms of these unique cellular phenotypes (ONL disorga-nization) and their impacts on the function and survival of photoreceptors and other retinal cell types over development.

Our study here also emphasizes the importance of tailoring gene therapy regimens to tackle individual pathogenic mechanisms. For instance, while supplementing WT CRX might be sufficient to rescue a hypomorphic/loss-of-function mutant, simultaneous elimination of a gain-of-function *CRX* product would be necessary to rescue dominant mutants, as exemplified in a recent report of allele-specific gene editing to rescue dominant CRX-associated LCA7 phenotypes in a retinal organoid model (*Chirco et al., 2021*). We believe that this principle also applies to other dominant neurological diseases. Additionally, with the refinement of the CRX mechanistic model, when new disease mutations are identified, genetic counsellors can now provide more informed predictions of disease progression and future visual deficits. This information is important for individuals to be psychologically prepared and seek necessary assistance to improve their quality of life.

# Materials and methods

**Key resources table**

| Reagent type (species) or resource | Designation | Source or reference | Identifiers | Additional information |
|---|---|---|---|---|
| Gene (*Homo sapiens*) | *CRX* | HGNC | HGNC:2383 | |
| Gene (*M. musculus*) | *Crx* | MGI | MGI:1194883 | |
| Strain, strain background (*Escherichia coli*) | BL21 (DE3) | MilliporeSigma | CMC0016 | Electrocompetent cells |
| Genetic reagent (*M. musculus*) | WT (C57BL/6J) | The Jackson Laboratory | Cat #000664 | |
| Genetic reagent (*M. musculus*) | *Crx*$^{E80A}$ (C57BL/6J) | This paper | | See Materials and methods |
| Genetic reagent (*M. musculus*) | *Crx*$^{K88N}$ (C57BL/6J) | This paper | | See Materials and methods |
| Genetic reagent (*M. musculus*) | *Crx*$^{R90W}$ (C57BL/6J) | *Tran et al., 2014* | | |
| Cell line (*Homo sapiens*) | HEK293T | ATCC | CRL-3216 | |
| Antibody | anti-CRX A-9 (mouse monoclonal) | Santa Cruz Biotechnology | sc-377138 | ChIP: 6 µg per 400 µl reaction |
| Antibody | anti-CRX M02 (mouse monoclonal) | Abnova Corp. | H00001406-M02 | WB: 1:1000 |
| Antibody | anti-HDAC1 H51 (rabbit polyclonal) | Santa Cruz Biotechnology | sc-7872 | WB: 1:1000 |
| Antibody | anti-mCAR (rabbit polyclonal) | MilliporeSigma | AB15282 | IH: 1:200 |
| Antibody | anti-RHO (mouse monoclonal) | MilliporeSigma | O4886 | IH: 1:200 |
| Antibody | Anti-RXRγ Y-20 (rabbit polyclonal) | Santa Cruz Biotechnology | sc-555 | IH: 1:100 |
| Antibody | IRDye 680RD Goat anti-Rabbit IgG Secondary Antibody (mouse polyclonal) | LI-COR | 926-68071 | WB: 1:10,000 |
| Antibody | IRDye 800CW Goat anti-Mouse IgG Secondary Antibody (mouse polyclonal) | LI-COR | 926-32210 | WB: 1:10,000 |
| Commercial assay or kit | Amicon Ultra-0.5 Centrifugal Filter Unit | Millipore | UFC500324 | |
| Commercial assay or kit | Dual-Luciferase Reporter Assay System | Promega | E1910 | |
| Commercial assay or kit | GST SpinTrap | Cytiva | 28952359 | |
| Commercial assay or kit | iScript Reverse Transcription Supermix | Bio-Rad Laboratories | 1708841 | |

*Continued on next page*

*Continued*

| Reagent type (species) or resource | Designation | Source or reference | Identifiers | Additional information |
|---|---|---|---|---|
| Commercial assay or kit | MEGAshortscript T7 Transcription Kit | Thermo Fisher Scientific | AM1354 | |
| Commercial assay or kit | MinElute PCR Purification Kit | QIAGEN | 28006 | |
| Commercial assay or kit | NE-PER Nuclear and Cytoplasmic Extraction Reagents | Thermo Scientific | 78833 | |
| Commercial assay or kit | Novex WedgeWell 12% Tris-Glycine Mini Protein Gels | Invitrogen | XP00122BOX | |
| Commercial assay or kit | NuPAGE 4 to 12%, Bis-Tris Mini Protein Gels | Invitrogen | NP0322BOX | |
| Commercial assay or kit | Phusion High-Fidelity PCR Master Mix with HF Buffer | New England Biolabs | M0531S | |
| Commercial assay or kit | SsoFast EvaGreen Supermix with Low ROX | Bio-Rad Laboratories | 1725211 | |
| Chemical compound, drug | Atropine sulfate solution | Bausch and Lomb | NDC 24208-825-55 | |
| Chemical compound, drug | Chameleon Duo Pre-stained Protein Ladder | LI-COR | 928-60000 | |
| Chemical compound, drug | Dithiothreitol (DTT) | Bio-Rad Laboratories | 1610611 | |
| Chemical compound, drug | Exonuclease I (*E. coli*) | New England Biolabs | M0293S | |
| Chemical compound, drug | Gibco Dulbecco's modified Eagle medium | Thermo Fisher Scientific | 11965084 | |
| Chemical compound, drug | Gibco fetal bovine serum | Thermo Fisher Scientific | 16000044 | |
| Chemical compound, drug | Glutathione Sepharose 4B resin | Cytiva | 17075601 | |
| Chemical compound, drug | Isopropyl-β-D-thiogalactopyranoside (IPTG) | Thermo Fisher Scientific | BP1755-10 | |
| Chemical compound, drug | Molecular Biology Grade Water | Corning | 46-000-CM | |
| Chemical compound, drug | Penicillin–streptomycin | Thermo Fisher Scientific | 15140122 | |
| Chemical compound, drug | Phosphate-buffered saline | Corning | 46-013-CM | |
| Chemical compound, drug | Roche cOmplete, Mini Protease Inhibitor Cocktail | MilliporeSigma | 11836153001 | |
| Chemical compound, drug | SeeBlue Plus2 Pre-stained Protein Standard | Invitrogen | LC5925 | |
| Chemical compound, drug | Triton X-100 | Sigma-Aldrich | T9284 | |
| Chemical compound, drug | TRIzol Reagent | Invitrogen | 15596026 | |
| Chemical compound, drug | VECTASHIELD HardSet Antifade Mounting Medium with DAPI | Vector Laboratories | H-1500-10 | |
| Software and algorithms | bedtools (v2.27.1) | *Quinlan and Hall, 2010* | | https://bedtools.readthedocs.io/en/latest/ |

*Continued on next page*

*Continued*

| Reagent type (species) or resource | Designation | Source or reference | Identifiers | Additional information |
|---|---|---|---|---|
| Software and algorithms | Bowtie2 (v 2.3.4.1) | *Langmead and Salzberg, 2012* | | https://bowtie-bio.sourceforge.net/bowtie2/index.shtml |
| Software and algorithms | BSgenome (v 1.58.0) | *Pagès, 2020* | | https://bioconductor.org/packages/BSgenome |
| Software and algorithms | Clustal Omega | *Goujon et al., 2010; Sievers et al., 2011* | | https://www.ebi.ac.uk/Tools/msa/clustalo/ |
| Software and algorithms | DAVID (v6.8) | *Sherman et al., 2022* | | https://david.ncifcrf.gov/ |
| Software and algorithms | deeptools (v3.0.0) | *Ramírez et al., 2016* | | https://deeptools.readthedocs.io/en/develop/ |
| Software and algorithms | DEseq2 (v1.30.1) | *Love et al., 2014* | | https://bioconductor.org/packages/DESeq2 |
| Software and algorithms | DiffBind (v3.0.15) | *Stark and Brown, 2012; Ross-Innes et al., 2012* | | https://bioconductor.org/packages/DiffBind |
| Software and algorithms | fastcluster (v1.1.26) | *Müllner, 2013* | | http://danifold.net/fastcluster.html |
| Software and algorithms | FastQC (v0.11.5) | *Andrews, 2010* | | http://www.bioinformatics.babraham.ac.uk/projects/fastqc/ |
| Software and algorithms | GraphPad Prism 8 | GraphPad Software | | https://www.graphpad.com/scientific-software/prism/ |
| Software and algorithms | GREAT (v4.0.4) | *McLean et al., 2010* | | http://great.stanford.edu/public/html/ |
| Software and algorithms | HOMER (v4.8) | Software and algorithms | | http://homer.ucsd.edu/homer/motif/ |
| Software and algorithms | IDR framework (v2.0.4) | *Li et al., 2011* | | https://github.com/nboley/idr; *Boley, 2017* |
| Software and algorithms | IGV Web App | *Robinson et al., 2011* | | https://igv.org/ |
| Software and algorithms | Jalview (v2.11.1.7) | *Waterhouse et al., 2009* | | https://www.jalview.org/ |
| Software and algorithms | kallisto (v0.46.2) | *Bray et al., 2016* | | https://github.com/pachterlab/kallisto; *Pachter Lab, 2023* |
| Software and algorithms | logomaker (v0.8) | *Tareen et al., 2020* | | https://logomaker.readthedocs.io/en/latest/ |
| Software and algorithms | MACS2 (v2.1.1.20160309) | *Zhang et al., 2008* | | https://github.com/macs3-project/MACS; *MACS3 project team, 2012* |
| Software and algorithms | matplotlib (v3.5.1) | *Hunter, 2007* | | https://matplotlib.org/ |
| Software and algorithms | MEME Suite (v5.0.4) | *Bailey et al., 2015* | | https://meme-suite.org/meme/index.html |
| Software and algorithms | rg.Mm.eg.db (v3.12.0) | *Carlson, 2019* | | https://bioconductor.org/packages/org.Mm.eg.db |
| Software and algorithms | pandas (v1.4.2) | *Reback et al., 2022* | | https://pandas.pydata.org/ |
| Software and algorithms | Picard (v2.21.4) | *Broad Institute, 2019* | | http://broadinstitute.github.io/picard/ |
| Software and algorithms | python (v3.9.12) | *Van Rossum and Fred, 1995* | | https://docs.python.org/3/reference/ |

*Continued on next page*

*Continued*

| Reagent type (species) or resource | Designation | Source or reference | Identifiers | Additional information |
|---|---|---|---|---|
| Software and algorithms | R (v4.0.3) | R Core Team, 2020 | | |
| Software and algorithms | rGREAT (v1.19.2) | *Gu et al., 2023* | | https://bioconductor.org/packages/rGREAT |
| Software and algorithms | samtools (v1.9) | *Li et al., 2009* | | http://www.htslib.org/ |
| Software and algorithms | scikit_posthocs (v0.7.0) | *Terpilowski, 2019* | | https://scikit-posthocs.readthedocs.io/en/latest/ |
| Software and algorithms | scipy (v1.8.1) | *Virtanen et al., 2020* | | https://scipy.org/ |
| Software and algorithms | seaborn (v0.11.2) | *Waskom, 2021* | | https://seaborn.pydata.org/ |
| Software and algorithms | Trim Galore (v0.6.1) | *Felix Krueger et al., 2023* | | https://github.com/FelixKrueger/TrimGalore/blob/master/Docs/Trim_Galore_User_Guide.md |
| Software and algorithms | tximport (v1.18.0) | *Soneson et al., 2015* | | https://bioconductor.org/packages/tximport |

## Resource availability

### Lead contact

Further information and requests for resources and reagents should be directed to and will be fulfilled by the lead contact Shiming Chen (chenshiming@wustl.edu).

### Materials availability

All unique/stable reagents generated in this study are available from the lead contact with a completed materials transfer agreement.

### Data and code availability

- The raw sequencing data and processed data generated in this study have been deposited at NCBI under the accession number GEO: GSE223659.
- Customized scripts and any additional information required to reproduce the analysis in this paper are available from GitHub at https://github.com/YiqiaoZHENG/CRXHD_mousemodel.

## Animal study and sample collection

### Mutation knock-in mouse model generation

CRISPR/Cas9-based genome editing was performed to generate the *Crx*[E80A] and *Crx*[K88N] mice as previously described (*Yang et al., 2014*). The Cas9 guide RNAs (gRNA) were designed based on proximity to the target amino acid and was synthesized using the MEGAshortscript T7 Transcription Kit (Thermo Fisher Scientific, Waltham, MA). The gRNAs were subsequently tested for cutting efficiency in cell culture by the Washington University Genome Engineering and iPSC Center. The validated gRNA and Cas9 protein were then microinjected into the pronuclei of C57Bl/6J- 0.5-dpc (days post coitum) zygotes along with the donor DNA, a 190-bp single-stranded oligodeoxynucleotide (ssODN) carrying either the *c.239A>G* substitution for p.E80A mutation or the *c.264G>T* substitution for p.K88N mutation (*Cho et al., 2009*). Embryos were then transferred into the oviduct of pseudo-pregnant female. Pups were generally delivered ~20 days after microinjection. Tissues from 10-day postnatal (P10) pups were collected by toe biopsy/tail for identification of the targeted allele by restriction digest (HinfI) of PCR amplified DNA first and then confirmed by Sanger sequencing (Genewiz).

Founders carrying the correct alleles were then bred with WT C57BL/6J mice (Jackson Laboratories, Bar Harbor, ME, Strain #000664) to confirm transmission. All experimental animals used were backcrossed at least 10 generations. Genotyping of mutation knock-in mice follows cycling conditions:

95°C for 2 min, 94°C for 30 s, 60°C for 30 s, 68°C for 60 s, repeat steps 2–4 for 34 cycles, 68°C for 7 min, and hold at 4°C. After PCR reaction, the amplified DNA fragments wee digested with HinfI. Sequences of gRNAs, ssODNs, and genotyping primers can be found in *Supplementary file 1c*. A representative DNA gel of HinfI digested genotyping DNA fragments can be found at *Figure 2— figure supplement 1B*.

## RNA-seq sample collection and library preparation

For each genotype, three biological replicates, two retinas per replicate from one male and one female mouse were analyzed. All retinas were processed for RNA simultaneously using TRIzol Reagent (Invitrogen, Waltham, MA) following the manufacturer's protocol. The quantity and quality of the RNA were assayed using Bioanalyzer (Agilent, Santa Clara, CA). Samples with a minimum RNA integrity number score of 8.0 were then selected for library construction as previously described (*Ruzycki et al., 2015*).

## Chromatin immunoprecipitation and library preparation

CRX chromatin immunoprecipitation was performed as previously published (*Chen et al., 2004*). Briefly, pooled nuclear extracts from six retinae were cross-linked with formaldehyde prior to immunoprecipitation with anti-CRX antibody A-9 (#sc-377138, Santa Cruz Biotechnology, Dallas, TX). Input controls were included as background. The libraries were prepared following the standard ChIP-seq protocol (*Schmidt et al., 2009*). The quantity and quality of the ChIP-seq libraries were assayed using Bioanalyzer (Agilent, Santa Clara, CA) prior to sequencing.

## ERG and statistical analyses

ERGs were performed on 1-month-old mice using UTAS-E3000 Visual Electrodiagnostic System (LKC Technologies Inc, MD). Mice were dark-adapted overnight prior to the tests. Mouse body temperature was kept at $37 \pm 0.5°C$ during the tests. Pupils were dilated with 1% atropine sulfate solution (Bausch and Lomb). Platinum 2.0 mm loop electrodes were placed on the cornea of each eye. A reference electrode was inserted under the skin of the mouse's head and a ground electrode was placed under the skin near mouse's tail. Retinal response to full-field light flashes (10 µs) of increasing intensity were recorded; maximum flash intensity for dark-adapted testing was 0.895 cd*s/m². Following dark-adapted tests, mice were light adapted under light condition (about 29.2 cd/mm) for 10 min and exposed to 10 µs light flashes of increasing intensity; maximum flash intensity for light-adapted testing was 2.672 cd*s/m². ERG responses of biological replicates were recorded, averaged, and analyzed using GraphPad Prism 8 (GraphPad Software, CA). The mean peak amplitudes of dark-adapted A- and B-waves and light-adapted B-waves were plotted against log values of light intensities (cd*s/m²). The statistics were obtained by two-way analysis of variance (ANOVA) with multiple pairwise comparisons (Tukey's).

## Histology and immunohistology chemistry

Enucleated eyes were fixed at 4°C overnight for paraffin-embedded sections. Each retinal cross-section was cut 5 µm thick on a microtome. H&E staining was performed to examine retinal morphology. For IHC staining, sections firstly went through antigen retrieval with citrate buffer, and blocked with a blocking buffer of 5% donkey serum, 1% bovine serum albumin, 0.1% Triton X-100 in 1× phosphate-buffered saline (PBS) (pH 7.4) for 1 hr. Sections were then incubated with primary antibodies at 4°C overnight. Sections were washed with 1× PBS containing 0.01% Triton X-100 (PBST) for 30 min, and then incubated with specific secondary antibodies for 1 hr. Primary and secondary antibodies were applied with optimal dilution ratios. All slides were mounted with VECTASHIELD HardSet Antifade Mounting Medium with DAPI (Vector Laboratories, Inc, CA). All images were taken on a Leica DB5500 microscope. All images were acquired at 1000 µm from ONH for ≥P21 samples and at 500 µm from ONH for P0, P3, and P10 samples.

## Biochemistry

### Protein expression

Expression plasmids for GST-WT, E80A, and R90W HDs were published previously (*Chen et al., 2002*). Plasmid for GST- K88N HD was generated by site-directed mutagenesis from the pGEX4T2-CRX WT HD backbone. In vivo protein expression and purification were done as previously described (*Chen et al., 2002*). Briefly, 0.05 mM isopropyl-β-D-thiogalactopyranoside was added to *E. coli* BL-21 (DE3) cell cultures containing different CRX HD constructs at $OD_{600}$ = 0.6. The cultures were incubated for 2 hr or until $OD_{600}$ = 2.0 at 34°C and the cells were collected by centrifugation at 6000 rpm and 4°C for 15 min. Cell pellets were resuspended in 1× PBS (Corning, Corning, NY) and then lysed by sonication. 5 mM dithiothreitol (DTT) (Bio-Rad Laboratories, Inc, Hercules, CA) and 1% Triton X-100 (MilliporeSigma, Burlington, MA) was then added, and the mixtures were incubated with gentle shaking at 4°C for 30 min to maximize protein extraction. The separation of proteins from the cellular debris were then performed by centrifugation at 15,000 rpm for 10 min and filtered through a 0.45-μm membrane. Glutathione Sepharose 4B resin (Cytiva, Marlborough, MA) was first equilibrated with PBS before adding to the supernatant. 5× Halt Protease Inhibitor Cocktail and phenylmethylsulfonyl fluoride (PMSF) was added to minimize degradation. The mixtures were incubated with gentle shaking at 4°C overnight before loading on GST Spintrap columns (Cytiva, Marlborough, MA). The peptides were eluted following the manufacturer's protocol and buffer exchanged into CRX-binding buffer (*Lee et al., 2010*) using Amicon centrifugal filters (MilliporeSigma, Burlington, MA). The protein stock was supplemented with 10% glycerol before aliquoted and stored at −80°C.

### Protein quantification and visualization

The size and integrity of purified GST-CRX HDs were visualized with a native 12% Tris-Glycine sodium dodecyl sulfate–polyacrylamide gel electrophoresis (SDS–PAGE) gel in the absence any reducing agent. Protein concentration was measured by NanoDrop One$^c$ Microvolume UV-Vis Spectrophotometers (Thermo Fisher Scientific, Waltham, MA) and calculated using the equation: $C = (1.55 * A_{280}) − (0.76 * A_{260})$, where $C$ is the concentration of the protein in mg/ml, $A_{280}$ and $A_{260}$ are the absorbance of protein samples at 280 and 260 nm, respectively (*Roy et al., 2017*). The protein concentrations obtained with this method were comparable with Bicinchoninic Acid (BCA) protein quantification assays.

### Spec-seq library synthesis and purification

Single-stranded Spec-seq library templates and IRDye 700-labeled reverse complement primers (*Supplementary file 1b*) were ordered directly from Integrated DNA Technologies (IDT, Coralville, Iowa). The synthesis and purification of the double-stranded libraries followed previously published protocols (*Zuo et al., 2017*; *Zuo and Stormo, 2014*; *Roy et al., 2017*). Briefly, 100 pmol of template oligos and 125 pmol IRDye 700-labeled reverse complement primer F1 were mixed in Phusion High-Fidelity PCR Master Mix (NEB, Ipswich, MA). A 15-s denaturing at 95°C following a 10-min extension at 52°C afforded duplex DNAs. Subsequently, the mixture was treated with 1 μl Exonuclease I (NEB, Ipswich, MA) to remove excess ssDNA. The libraries were purified by MinElute PCR Purification Kit (QIAGEN, Hilden, Germany) and eluted in molecular biology graded water (Corning, Corning, NY).

### EMSA and sample preparation for sequencing

The protein–DNA-binding reactions was done in 1× CRX-binding buffer (60 mM KCl, 25 mM (4-(2-hyd roxyethyl)-1-piperazineethanesulfonic acid) HEPES, 5% glycerol, 1 mM DTT) (*Lee et al., 2010*). A fixed amount (*Supplementary file 1b*) of IRDye-labeled DNA libraries were incubated on ice for 30 min with varying concentrations of WT or mutant peptides in 20 μl reaction volume. The reaction mixtures were run at 4°C in native 12% Tris-Glycine PAGE gel (Invitrogen, Waltham, MA) at 160 V for 40 min. The IRDye-labeled DNA fragments in the bound and unbound fractions were visualized by Odyssey CLx and Fc Imaging Systems (LI-COR, Inc, Lincoln, NE). The visible bands were excised from the gels and DNAs were extracted with acrylamide extraction buffer (100 mM $NH_4OAc$, 10 mM $Mg(OAc)_2$, 0.1% SDS) then purified with MinElute PCR Purification Kit (QIAGEN, Hilden, Germany). The DNAs were amplified, barcoded by indexed Illumina primers. All indexed libraries were then pooled and

sequenced on a single 1 × 50 bp Miseq run at DNA Sequencing Innovation Lab at the Center for Genome Sciences & Systems Biology (CGS&SB, WashU).

## qRT-PCR

For each replicate, RNA from two retinae of a mouse was extracted using the NucleoSpin RNA kits (Takara Bio USA, Inc, San Jose, CA). RNA sample concentration and quality were determined with NanoDrop One$^c$ Microvolume UV-Vis Spectrophotometers (Thermo Fisher Scientific, Waltham, MA). 1 µg of RNA was used for cDNA synthesis with iScript cDNA Synthesis Kits (Bio-Rad, Hercules, CA) in a 20-µl reaction volume. Primers used in this study are listed in *Supplementary file 1b*. qRT-PCR reactions were assembled using SsoFast EvaGreen Supermix with Low ROX (Bio-Rad Laboratories, Inc, Hercules, CA) following the manufacturer's protocol. Data were obtained from Bio-Rad CFX96 Thermal Cycler following a three-step protocol: 1 cycle of 95°C 3 min, 40 cycles of 95°C 10 s, and 60°C 30 s. Data were exported and further processed with customized python script.

## Western blot

Experiments were performed using two biological replicates with two retinas for each replicate. Nuclear extracts were prepared using the NE-PER Nuclear and Cytoplasmic Extraction Reagents (Thermo Fisher Scientific, Waltham, MA) following manufacturer's instructions. 1× Roche cOmplete Mini Protease Inhibitor Cocktail (MilliporeSigma, Burlington, MA) was supplemented in all extraction reagents. 5 mM of DTT was added immediately before sample denaturing and protein was separated by running on Invitrogen NuPAGE Novex 4–12% Bis-Tris MiniGels (Invitrogen, Waltham, MA). Membrane transfer was done with the Blot mini blot module (Invitrogen, Waltham, MA) following the manufacturer's protocol. Membrane was probed with mouse monoclonal anti-CRX antibody M02 (1:1000, Abnova Corp, Taipei City, Taiwan) and rabbit polyclonal anti-HDAC1 antibody H51 (1:1000, Santa Cruz Biotechnology, Dallas, TX), visualized with IRDye 680RD goat anti-rabbit IgG and IRDye 800CW goat anti-mouse IgG secondary antibodies (1:10,000, LI-COR, Inc, Lincoln, NE). The membrane was then imaged using Odyssey CLx and Fc Imaging Systems (LI-COR, Inc, Lincoln, NE).

## Cell line transient transfection luciferase reporter assays

HEK293T cells (CRL-3216) were obtained directly from ATCC (American Type Culture Collection). The cells were used within 1 year of purchase and tested negative for mycoplasma contamination. Cells are cultured in Dulbecco's modified Eagle medium supplemented with 10% fetal bovine serum and penicillin–streptomycin following the manufacturer's protocol. Cells were transfected with calcium phosphate transfection protocol in 6-well plates as previously described (*Chen et al., 2002*; *Tran et al., 2014*). Experimental plasmids and usage amount are described in *Supplementary file 1b*. Typically, 48 hr after transfection, cells were harvested, digested, and assayed for luciferase activity using Dual-Luciferase Reporter Assay System (Promega, Madison, WI) following the manufacturer's protocol. Data were collected using TD-20/20 Luminometer (Turner Designs, East Lyme, CT) and further processed with customized python scripts.

## Data analysis

### HD sequence alignment

The full-length protein sequences for the selected TFs were first aligned with Clustal Omega (EMBL-EBI, UK). Aligned sequences of the third HD helix were then extracted to generate *Figure 1B* using Jalview (v2.11.1.7). A list of the accession numbers for the selected TFs can be found in *Supplementary file 1a*.

### Determination of relative binding affinity with Spec-seq

For a biomolecular interaction between a protein $P$ and a particular DNA sequence, $S_i$, the interaction can be diagrammed as:

$$P + S_i \rightleftharpoons P \cdot S_i \tag{1}$$

where $P \cdot S_i$ refers to the protein–DNA complex. The affinity of the protein $P$ to sequence $S_i$ is defined as the association constant $K_A$, or its reciprocal, the dissociation constant $K_D$. The $K_A$ of the

protein–DNA interaction is determined by measuring the equilibrium concentrations of each reactant and the complex:

$$K_A\left(S_i\right) = \frac{[P \cdot S_i]}{[P] \cdot [S_i]} \tag{2}$$

where [...] refers to concentrations. As in a typical Spec-seq experiment, thousands of different DNA sequences compete for the same pool of proteins, their relative binding affinities (the ratio of their $K_A$) can be determined by measuring the concentrations of each sequence in the bound and unbound fractions without measuring the free protein concentrations, which is often the most difficult to measure accurately:

$$K_A\left(S_1\right) : K_A\left(S_2\right) : \ldots : K_A\left(S_n\right) = \frac{[P \cdot S_1]}{[S_1]} : \frac{[P \cdot S_2]}{[S_2]} : \ldots : \frac{[P \cdot S_n]}{[S_n]} \tag{3}$$

In a binding reaction involving TF and a library of DNAs, the concentration of bound and unbound species are directly proportional to the number of individual DNA molecules in each fraction which can be obtained directly from sequencing data. With enough counts in each fraction, we can accurately estimate the ratios of concentrations from counts with the relationship:

$$\frac{[P \cdot S_i]}{[P \cdot S_j]} \approx \frac{N_B\left(S_i\right)}{N_B\left(S_j\right)} \text{ and } \frac{[S_i]}{[S_j]} \approx \frac{N_U\left(S_i\right)}{N_U\left(S_j\right)} \tag{4}$$

where $N_U$ denotes counts in the unbound fraction and $N_B$ denotes counts in the bound fraction. Therefore, the binding affinity of a sequence variant $S_x$ relative to the reference sequence $S_{ref}$ can be calculated by:

$$\frac{K_A\left(S_x\right)}{K_A\left(S_{ref}\right)} \approx \frac{[P \cdot S_x]}{[P \cdot S_{ref}]} \frac{[S_{ref}]}{[S_x]} \approx \frac{N_B\left(S_x\right)}{N_B\left(S_{ref}\right)} \frac{N_U\left(S_{ref}\right)}{N_U\left(S_x\right)} \tag{5}$$

The natural logarithms of these ratios are the relative binding free energies in the units of kcal/mol. The relative free energy of the reference site for each CRX HD was set to 0.

## Spec-seq data analysis and energy logo visualization

The sequencing results were first filtered and sorted based on conserved regions and barcodes. Reads with any mismatch in the conserved regions were discarded prior to further analysis as described previously (**Stormo et al., 2015**; **Zuo and Stormo, 2014**; **Roy et al., 2017**). The ratio of individual sequence in bound and unbound reads was calculated as a measurement of relative binding affinity (**Equation 5**Equation 5) compared to the consensus sequence. The relative binding energy was then derived from the natural logarithm of the relative binding affinity and represented in kcal/mol units.

For WT CRX HD and all the mutants, the energy weight matrices (ePWMs) were generated based on the regression of the TF's binding energy to its reference sequence. Only sequences within two mismatches to the reference were used to generate the ePWMs. Energy logos were generated from ePWMs after normalizing the sum of energy on each position to 0 and the negative energy values were plotted such that preferred bases are on top. The sequence logos were generated from ePWMs with python package logomaker (v0.8). The ePWMs for all CRX HDs are listed in **Supplementary file 1c**.

## ChIP-seq data analysis

$2 \times 150$ bp reads from Illumina NovaSeq were obtained for all samples with a minimum depth of 22 M reads at Novogene (Beijing, China). For each sample, reads from two sequencing lanes were first concatenated and run through Trim Galore (v0.6.1) (**Felix Krueger et al., 2023**) to remove adapter sequences and then QC by FastQC (v0.11.5) (**Andrews, 2010**). The trimmed reads were then mapped to the mm10 genome using Bowtie2 (v 2.3.4.1) (**Langmead and Salzberg, 2012**) with parameters -X 2000 --very-sensitive. Only uniquely mapped and properly paired reads were retained with samtools (v1.9) (**Li et al., 2009**) with parameters -f 0x2 -q 30. Mitochondria reads were removed with samtools (v1.9) (**Li et al., 2009**). Duplicated reads were marked and removed with Picard (v2.21.4) (**Broad**

*Institute, 2019*). Last, reads mapped to the mm10 blacklist regions were removed by bedtools (v2.27.1) (*Quinlan and Hall, 2010*) by intersect -v. bigWig files were generated with deeptools (v3.0.0) (*Ramírez et al., 2016*) with command bamCoverage --binSize 10 -e --normalizeUsing CPM and visualized on IGV Web App (*Robinson et al., 2011*). For each genotype, an average binding intensity bigWig file from two replicates was generated with deeptools (v3.0.0) (*Ramírez et al., 2016*) command bamCompare –operation mean with default parameters.

Peak-calling was done with MACS2 (v2.1.1.20160309) (*Zhang et al., 2008*) on individual replicate with the default parameters. For each genotype, we then generated a genotype-specific high confidence peakset by intersection of peaks called in two replicates. IDR framework (v2.0.4) (*Li et al., 2011*) were used to generate quality metrics for the processed ChIPseq data. R package DiffBind (v3.0.15) (*Stark and Brown, 2012*; *Ross-Innes et al., 2012*) and DEseq2 (v1.30.1) (*Love et al., 2014*) were then used to re-center peaks to ±200 bp regions surrounding summit, generate normalized binding intensity matrix, and differential binding matrix. We defined differentially bound peaks between each mutant and WT sample if the absolute $log_2FC$ is more than 1.0, corresponding to twofold, and the false discovery rate (FDR) is smaller than 5e−2.

To associate peaks to genes, we used Genomic Regions Enrichment of Annotations Tool (GREAT v4.0.4) (*McLean et al., 2010*) through the R package rGREAT (v1.19.2) (*Gu et al., 2023*). Each peak was assigned to the closest TSS within 100 kb.

## Binding intensity heatmap and clustering

To generate the binding intensity heatmap in *Figure 2A*, we first compiled the genotype-specific high confidence peakset for all genotypes into a single consensus peakset and only peaks with at least 5 cpm in all genotypes were retained. Python package fastcluster (v1.1.26) (*Müllner, 2013*) was used to perform hierarchical clustering of the consensus peakset intensity matrix with parameters method='single', metric='euclidean'. The genomic regions corresponding to the two major clusters were exported and used to generate binding intensity heatmaps with deeptools (v3.0.0) (*Ramírez et al., 2016*).

## Genomic region enrichment of CRX peaks

Peak annotation in *Figure 2D* were obtained using annotatePeaks.pl from HOMER (v4.8).

## De novo motif searching

The mm10 fasta sequences for each genotype-specific peaks were obtained using R package BSgenome (v 1.58.0) (*Pagès, 2020*). De novo motif enrichment analysis for each set of sequences was then performed with MEME-ChIP in MEME Suite (v5.0.4) (*Bailey et al., 2015*) using order 1 Markov background model and default parameters. Since HD motifs are relatively short and can be repetitive (e.g., K88N motif), we reported DREME (*Bailey, 2011*) found motifs for *Figure 2*, which is more sensitive than MEME to find short, repetitive motifs.

## RNA-seq data analysis

2 × 150 bp reads from Illumina NovaSeq were obtained for all samples with a minimum depth of 17 M reads at Novogene (Beijing, China). Sequencing reads were first run through Trim Galore (v0.6.1) (*Felix Krueger et al., 2023*) to remove adapter sequences and then QC by FastQC (v0.11.5) (*Andrews, 2010*). Trimmed reads were then mapped to the mm10 genome and quantified with kallisto (v0.46.2) (*Bray et al., 2016*). Kallisto output transcript-level abundance matrices were then imported and summarized into gene-level matrices with R package tximport (v1.18.0) (*Soneson et al., 2015*). DEseq2 (v1.30.1) (*Love et al., 2014*) was then used for normalization and differential expression analysis. The normalized count and differential expression matrices were then exported and further processed with customized python scripts.

We defined differentially expressed genes between each mutant and WT sample if the absolute $log_2FC$ is more than 1.0, corresponding to twofold, and the FDR is smaller than 1e−2. For comparison between heterozygous and homozygous mutants, we first filtered genes with at least 5 cpm and then those that were called differentially expressed compared with WT in at least one mutant genotype.

We retrieved gene names in *Supplementary file 1e–g* from the Database for Annotation, Visualization and Integrated Discovery (DAVID, v6.8) (*Sherman et al., 2022*).

## Definition of CRX-dependent and -independent gene set

We first identified CRX peaks that were bound in the WT sample but lost in *R90W/W* sample (log$_2$FC <−1 and FDR <5e−2). This yielded a total of 7677 peaks. We then found the genes associated with these peaks. We defined a gene to be CRX-dependent activated if its expression was down in adult (P21) *R90W/W* RNA-seq sample (log$_2$FC <−0.6 and FDR <1e−5). Similarly, a gene is defined as CRX-dependent suppressed if its expression was up in adult *R90W/W* RNA-seq sample (log$_2$FC >0.6 and FDR <1e−5). A gene is defined as CRX independent if its expression was not significantly affected in adult *R90W/W* RNA-seq samples. There were 617 CRX-dependent activated, 135 CRX-dependent suppressed, and 5565 CRX-independent genes. Manual inspection of the CRX-dependent suppressed genes revealed no clear association with photoreceptor development. Therefore, we did not further pursue this gene set. The complete list of CRX-dependent activated genes that showed differential expression in at least one of the HD mutant retinas can be found in *Supplementary file 1e*. The lists for CRX-independent genes that showed differential expression in *Crx^{E80A}* or *Crx^{K88N}* mutant retinas can be found in *Supplementary file 1f, g*, respectively.

## GO analysis

GO analysis in *Figure 3—figure supplement 1B, E* and *Figure 3—figure supplement 2F* was performed using R package clusterProfiler (v4.0.5) (*Yu et al., 2012*; *Wu et al., 2021*) with the genome-wide annotation package org.Mm.eg.db (v3.12.0) (*Carlson, 2019*). Redundant enriched GO terms were removed using simplify() function with parameters cutoff = 0.7, by="p.adjust". The enrichment analysis results were then exported in table format and further processed for plotting with python.

## Aldiri et al. RNA-seq data re-analysis

The RNA-seq data from *Aldiri et al., 2017* were obtained from GEO under accession numbers GSE87064. The reads were processed similarly as all other RNA-seq data generated in this study. For *Figure 3—figure supplement 1C, F*, *Figure 3—figure supplement 2C, E*, *Figure 4—figure supplement 1A, B*, expression row *z*-scores were calculated using average cpm from replicates at each age.

## Statistical analysis

One-way ANOVA with Turkey honestly significant difference test in *Figure 6A*, *Figure 2—figure supplement 1C*, and *Figure 6—figure supplement 1D* was performed with python packages scipy (v1.8.1) (*Virtanen et al., 2020*) and scikit_posthocs (v0.7.0) (*Terpilowski, 2019*). Two-sided Mann–Whitney *U*-test in *Figure 3—figure supplement 2A* was performed with python package scipy (v1.8.1) (*Virtanen et al., 2020*).

# Acknowledgements

We thank Mingyan Yang and Guangyi Ling for technical assistance, Susan Penrose and Mike Casey from the Molecular Genetics Service Core for generating E80A and K88N mutation knock-in mice lines, Inez Oh for RNA-seq sample collection and processing, and J Hoisington-Lopez and M Crosby from DNA Sequencing Innovation Lab at the Center for Genome Sciences & Systems Biology for sequencing assistance. This work was supported by NIH grants EY012543 (to S Chen), EY032136 (to S Chen), EY002687 (to WU-DOVS), and the Stein Innovation Award (to SC) and unrestricted funds (to WU-DOVS) from Research to Prevent Blindness. We also thank Mr. Artur Widlak for the generous gift from Widłak Family CRX Research Fund.

## Additional information

### Funding

| Funder | Grant reference number | Author |
|---|---|---|
| National Institute of Health | EY012543 | Shiming Chen |
| National Institute of Health | EY032136 | Shiming Chen |
| Research to Prevent Blindness | Stein Innovation Award | Shiming Chen |
| Widłak Family | CRX Research Fund | Shiming Chen |

The funders had no role in study design, data collection and interpretation, or the decision to submit the work for publication.

### Author contributions

Yiqiao Zheng, Conceptualization, Visualization, Methodology, Writing – original draft, Writing – review and editing, Performed Spec-seq and luciferase experiments; formal analysis of Spec-seq, CRX ChIP-seq, RNA-seq and luciferase data; Chi Sun, Conceptualization, Visualization, Methodology, Writing – review and editing, Performed immunochemistry and ERG experiments; analyzed immunochemistry and ERG data; Xiaodong Zhang, Investigation, Performed CRX ChIP-seq experiments; Philip A Ruzycki, Conceptualization, Methodology, Writing – review and editing, Assisted in CRX ChIP-seq and RNA-seq data analysis; Shiming Chen, Conceptualization, Resources, Supervision, Funding acquisition, Methodology, Writing – review and editing

### Author ORCIDs

Yiqiao Zheng ⓘ https://orcid.org/0000-0003-4133-0439
Shiming Chen ⓘ http://orcid.org/0000-0002-9299-5767

### Ethics

All procedures involving mice were approved by the Animal Studies Committee of Washington University in St. Louis and performed under Protocol 21-0414 (to SC). Experiments were carried out in strict accordance with recommendations in the Guide for the Care and Use of Laboratory Animals of the National Institutes of Health (Bethesda, MD), the Washington University Policy on the Use of Animals in Research; and the Guidelines for the Use of Animals in Visual Research of the Association for Research in Ophthalmology and Visual Sciences. Every effort was made to minimize the animals' suffering, anxiety, and discomfort.

Reviewer #1 (Public Review): https://doi.org/10.7554/eLife.87147.4.sa1
Reviewer #2 (Public Review): https://doi.org/10.7554/eLife.87147.4.sa2
Author Response: https://doi.org/10.7554/eLife.87147.4.sa3

## Additional files

### Supplementary files

• Supplementary file 1. Supplementary materials. (a) Homeodomain transcription factor (HD TF) accession numbers (related to *Figure 1B*). (b) Plasmids and primers for biochemistry experiments. (c) Spec-seq ePWMs (related to *Figure 1G, K–M*, and *Figure 1—figure supplement 2I–L*). (d) DNA sequences for mutation knock-in mice generation. (e) CRX-dependent activated genes differentially expressed in at least one mutant mouse model (ordered as in *Figure 3B*). (f) CRX-independent genes mis-regulated in *Crx^{E80A}* mutants (ordered as in *Figure 3—figure supplement 2C*). (g) CRX-independent genes mis-regulated in *Crx^{K88N}* mutants (ordered as in *Figure 3—figure supplement 2E*). (h) Annotation for phototransduction genes in *Figure 4B*. (i) Quantification and statistical analysis (related *Figure 2—figure supplement 1C* and *Figure 3—figure supplement 2A*)

• MDAR checklist

## Data availability

The raw sequencing data and processed data generated in this study have been deposited at NCBI under the accession number GEO: GSE223659. Customized scripts and any additional information required to reproduce the analysis in this paper are available from GitHub at https://github.com/YiqiaoZHENG/CRXHD_mousemodel copy archived at *Zheng, 2023*.

The following dataset was generated:

| Author(s) | Year | Dataset title | Dataset URL | Database and Identifier |
|---|---|---|---|---|
| Zheng Y, Sun C, Zhang X, Ruzycki P, Chen S | 2023 | Missense mutations in CRX homeodomain cause dominant retinopathies through two distinct mechanisms | https://www.ncbi.nlm.nih.gov/geo/query/acc.cgi?&acc=GSE223659 | NCBI Gene Expression Omnibus, GSE223659 |

The following previously published dataset was used:

| Author(s) | Year | Dataset title | Dataset URL | Database and Identifier |
|---|---|---|---|---|
| Aldiri I, Xu B, Wang L, Chen X | 2017 | The Dynamic Epigenetic Landscape of the Retina During Development, Reprogramming, and Tumorigenesis | https://www.ncbi.nlm.nih.gov/geo/query/acc.cgi?acc=GSE87064 | NCBI Gene Expression Omnibus, GSE87064 |

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
