## [Editor Report · eLife assessment]

This manuscript will be of interest to readers in the field of neural development and neurodegeneration. The study is **important** as it examines two disease-causing mutations within the homeodomain transcription factor Cone-Rod Homeobox (CRX) that causes retinopathy in humans. The data are **solid**, and the work contributes to our understanding of the underlying pathogenetic mechanisms.

---

## [Referee Report · Reviewer #1 (Public Review)]

The manuscript by Zheng et al. examined the disease-causing mechanisms of two missense mutations within the homeodomain (HD) of CRX protein. Both mutations were found in humans and can produce severe dominant retinopathy. The authors investigated the two CRX HD mutants via in vitro DNA-binding assay (Spec-seq), in vivo chromatin-binding assay (ChIP-seq), in vivo expression assay of downstream target genes (RNA-seq), and retinal histological and functional assays. They concluded that p.E80A increased the transactivation activity of CRX and resulted in precocious photoreceptor differentiation, whereas p.K88N significantly changed the binding specificity of CRX and led to defects in photoreceptor differentiation and maintenance. The authors performed a significant amount of analyses. The claims are sufficiently supported by the data. The results not only uncovered the underlying disease-causing mechanisms, but also can significantly improve our understanding of the interaction between HD-TF and DNA during development.

---

## [Referee Report · Reviewer #2 (Public Review)]

Zheng et al., investigated the molecular and functional mechanisms of two homeodomain missense mutations causing human retinal photoreceptor degeneration diseases in photoreceptor development regulated by the CRX transcription factor. They analyzed the E80A mutation associated with dominant cone-rod dystrophy (CRD) and the K88N mutation associated with dominant Leber Congenital Amaurosis (LCA). The authors found that E80A CRX binds to the same target DNA sites as WT CRX, but the binding specificity of K88N CRX is altered from that of WT in an in vitro assay. They generated Crx(E80A) and Crx(K88N) KI mice and performed ChIP assay and observed that K88N CRX binds to novel genomic regions from the WT-binding sites, while E80A binds to the WT sites. In addition, using the KI mice, they found that E80A and K88N differently affect the expression of Crx target genes. The authors may want to provide explicit clarification on whether CRX E80A mice exhibit cone development and/or degeneration defects.

This study is well executed with proper and solid methodologies, and the manuscript is clearly written. This study gives us the insights into how single missense CRX mutations lead to different types of human retinal photoreceptor degeneration diseases.

---

## [Author Response]

The following is the authors’ response to the previous reviews

Thank you for sending our revised manuscript for review and the positive editorial comments. On behalf of all authors, I would like to, again, thank the reviewers for their critical reading of our revised manuscript and for providing further suggestions. We have revised the introduction and discussion sections to specifically address the comments made by Reviewer #2. Please see below for detailed explanations.

**Reviewer #2 (Public Review):**
Overall, the authors have significantly improved the manuscript, but there is still an unclarified point. In response to the inquiry in the initial review on how extent E80A KI mice function as a pathological model of dominant CoRD, the authors add data (Figures S7) and described the sixth section in the discussion. However, the authors mentioned that it is technically too challenging because of a small number of cones. The point is not clear to me, but it is possible to analyze cone differentiation and degeneration by immunostaining at multiple stages even though cone number is small. Cone arrestin and S- and M-opsins become positive at early postnatal stages in the mouse retina. Cone arrestin seems earlier than cone opsins. Cones seem born by detecting RXRg at P0, but are cone arrestin and/or cone opsins expressed in early postnatal E80A/+ retina? If positive, how about an apoptosis marker? If negative, it seems to be a cone development phenotype rather than cone degeneration phenotype. If so, authors should modify the expression to say that the E80A retina underlies CoRD-like phenotype. It seems an overstatement.

We greatly appreciate Reviewer 2’s suggestions on further investigating cone photoreceptor phenotypes in the CRX E80A KI mouse model. All the points raised deserve a comprehensive and in-depth study. However, the focus of the current manuscript is to establish a general framework for understanding different missense mutations in homeodomain TFs beyond CRX. We believe that a separate and dedicated study is more appropriate to detail the quantitative molecular and cellular mechanisms of CRX E80A dysfunction in cone and rod photoreceptors, as stated in the last sentence of discussion section paragraph 6: “… quantitative characterization of CRX E80A molecular functions in a cone dominant retina warrants further study to understand its selective effect on the cone differentiation program and help elucidate WT CRX regulatory principles in early photoreceptor development.”.

Clinical diagnosis of cone-rod dystrophy (CoRD) is largely based on functional deficits of cones and rods. 1-month electroretinogram (ERG) (Figures 5K-M) shows no cone-mediated light responses and reduced rod functions in CrxE80A/+ mouse. These ERG deficits in the CRX E80A KI mouse model are in agreement with CoRD characteristics. Thus, it is reasonable to say that CRX E80A KI retina phenotype resembles CoRD phenotype.

**Reviewer #2 (Recommendations For The Authors):**
As a minor comment, in page 8, second section, "Previous studies have demonstrated the CRX is activated shortly after cell cycle exit in retinal progenitor cells fated to be photoreceptor.", the authors cited refs 66 and 67, which were in 2105 and 2016. However, it was demonstrated in the paper of J. Neurosci.31(46), 16792-807, 2011, Figure 1. The authors need to be scientifically fair to cite the JN 2011 paper.

In response to this comment above, the authors cited the JN 2011 paper in a modified sentence of "Animal studies have demonstrated that Crx is first expressed in post-mitotic photoreceptor precursors and maintained throughout life (Refs.13-15)", moved from the discussion to the introduction. To my knowledge, the JN2011 (new Ref 15) is the first study directly demonstrated that Crx begins to be expressed shortly after cell cycle exit of retinal progenitor cells. Refs. 13 and 14 showed Crx expression in adult stage photoreceptors but did not directly demonstrate the Crx expression in post-mitotic photoreceptor precursors. To be scientifically precise, the references should be cited as "Animal studies have demonstrated that Crx is first expressed in post-mitotic photoreceptor precursors (Ref. 15) and maintained throughout life (Refs.13 and 14)".”Thanks to the reviewer for the precise instruction. We have adjusted the reference order as follows: “Animal studies have demonstrated that Crx is first expressed in post-mitotic photoreceptor precursors13 and maintained throughout life14,15.”, where JN2011 paper is reference 13.